# Health system strengthening in fragile and conflict-affected states: A review of systematic reviews

Birke Bogale[1,2]*, Sasha Scambler[1], Aina Najwa Mohd Khairuddin[1,3], Jennifer E. Gallagher[1]

1 Faculty of Dentistry, Oral & Craniofacial Sciences, King's College London, London, United Kingdom, 2 Department of Dental and Maxillofacial Surgery, St. Paul's Hospital Millennium Medical College, Addis Ababa, Ethiopia, 3 Department of Community Oral Health and Clinical Prevention, Faculty of Dentistry, Universiti Malaya, Kuala Lumpur, Malaysia

* birke.bogale@kcl.ac.uk

**Data Availability Statement:** We reviewed published English language literature, and all the

## Abstract

### Background

Globally, there is growing attention towards health system strengthening, and the importance of quality in health systems. However, fragile and conflict-affected states present particular challenges. The aim of this study was to explore health system strengthening in fragile and conflict-affected states by synthesising the evidence from published literature.

### Methods

We conducted a review of systematic reviews (Prospero Registration Number: CRD42022371955) by searching Ovid (Medline, Embase, and Global Health), Scopus, Web of Science, and the Cochrane Library databases. Only English-language publications were considered. The Joanna Briggs Institute (JBI) Critical Appraisal Tool was employed to assess methodological quality of the included studies. The findings were narratively synthesised and presented in line with the Lancet's 'high-quality health system framework'.

### Results

Twenty-seven systematic reviews, out of 2,704 identified records, considered key dimensions of health systems in fragile and conflict-affected states, with the 'foundations' domain having most evidence. Significant challenges to health system strengthening, including the flight of human capital due to safety concerns and difficult working conditions, as well as limited training capacities and resources, were identified. Facilitators included community involvement, support systems and innovative financing mechanisms. The importance of coordinated and integrated responses tailored to the context and stage of the crisis situation was emphasised in order to strengthen fragile health systems. Overall, health system strengthening initiatives included policies encouraging the return and integration of displaced healthcare workers, building local healthcare workers capacity, strengthening

details and summaries are included within the
manuscript, and supporting documents.

**Funding:** BB is funded by the King's College
London 'Africa International PGR Scholarships
2021-22' (URL: https://www.kcl.ac.uk/study-
legacy/funding/africa-international-pgr-
scholarships) to support her study 'Dental and
Health Sciences Research MPhil/PhD'. JEG and SS
are salaried by King's College London. However,
there is no funding specific to this research paper.

**Competing interests:** The authors have declared
that no competing interests exist.

education and training, integrating healthcare services, trust-building, supportive supervision, and e-Health utilisation.

## Conclusion

The emerging body of evidence on health system strengthening in fragile and conflict-affected states highlights its complexity. The findings underscore the significance of adopting a comprehensive approach and engaging various stakeholders in a coordinated manner considering the stage and context of the situation.

## Introduction

Fragile and conflict-affected states (FCAS) encompass countries and territories classified by the World Bank, based on specific criteria that identify regions with significant levels of institutional and social fragility, as well as those affected by violent conflict [1]. They are home to about one billion people across more than 40 countries [2]. However, approximately one-quarter of the global population, or two billion individuals, reside in conflict-affected areas with millions forcibly displaced due to active conflicts, violence, and human rights violations [3]. The term 'fragile states' has been defined and indexed differently by various international development and aid organisations based on their own contextual considerations and underlying political motives [4]. Currently, alternative terms such as 'situations' or 'contexts' are increasingly being used instead of 'states' due to the recognition that fragility can have implications for all aspects of life, down to an individual level [1, 4–6]. 'Conflict-affected' refers to situations where existing challenges are caused by ongoing or past conflicts [6]. While fragility has often been associated with armed conflicts, particularly in low- and middle-income countries (LMICs), it is a multidimensional and complex issue that can manifest in any setting or system, including the most developed countries [7, 8]. This has been observed, for example, with natural disasters, political instabilities, and civil unrest, as well as the COVID-19 pandemic [5, 9, 10]. Fragile states are highly susceptible to humanitarian crises, which can be either total or partial in their impact on the system; they are also vulnerable to domestic and international conflicts and shocks [11]. Often, humanitarian crises precede armed conflicts, epidemics, famines, natural disasters, and other major emergencies [12]. Moreover, whilst the list of FCAS is updated annually, some countries remain regularly classified as such [1, 5].

The challenges faced by health systems in FCAS are complex. Basic health system functions are distorted, access to essential healthcare is minimal, and health outcomes are poor [9, 13–17]. FCAS have the lowest health indicators and the weakest health systems [18]. Consequently, life expectancy in these regions is significantly lower than the global average [14]. Besides, armed conflicts are considered global health problems as they give rise to new 'disease burdens', whilst existing healthcare needs persist [19, 20]. The limitation of access and continuity of care may also contribute to the increased burden of diseases, particularly chronic and non-communicable diseases (NCDs), and associated morbidity and mortality, especially in areas affected by protracted and recurrent crises [19–22]. Often, unregulated local health providers dominate the healthcare services in fragile states, with minimal interaction with formal public or private health sectors [20]. This situation arises due to the disruption and weakening of public healthcare systems, resource depletion, governance breakdown, and service cessation [13, 15]. Hence, various 'actors' such as United Nations agencies, the International Red Cross and Red Crescent Movement, non-governmental organisations (NGOs), civil society groups,

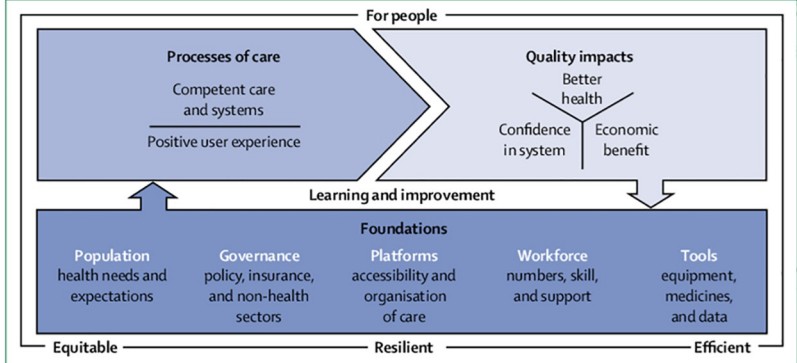

**Fig 1. High-quality health system framework. Source:** Kruk ME, Gage AD, Arsenault C, Jordan K, Leslie HH, Roder-DeWan S, et al. High-quality health systems in the Sustainable Development Goals era: time for a revolution. Lancet Glob Health. 2018;6(11): e1196-e252.

private healthcare providers, and traditional medicine practitioners partially meet the health-care needs of the population [20, 23].

In recent years, there has been increasing attention given to health system strengthening (HSS) [13, 24–26]. Health has become a prominent area of discussion and diplomatic engagement worldwide, with the involvement of numerous NGOs and aid agencies contributing to strengthening health systems and improving global health outcomes [27, 28]. Consequently, scholars and organisations, including the World Health Organisation (WHO), have proposed several HSS strategies and frameworks [29]. However, in the past, many of these frameworks did not adequately consider the aspect of quality; and there is now growing emphasis on its importance in health systems [26, 30]. On the basis of this, the Lancet Commission in 2018 proposed a 'high-quality health system framework', based on existing frameworks and definitions [30]. It encompasses three key domains: Foundations, Processes of care, and Quality impacts, each with its own distinct components (Fig 1). This framework serves as a foundation for our review.

Overall, the objective of HSS is to restore health systems to their previous state or create more modern and resilient systems. However, the unpredictable nature of crisis situations complicates and slows down the process due to unforeseen setbacks and challenges [15, 31]. Nevertheless, HSS in FCAS has garnered considerable attention in the academic sphere, with numerous articles, including systematic reviews, focusing on various aspects of health systems. Therefore, this study reviewed systematic reviews of HSS, with the aim of exploring HSS in FCAS by synthesising the evidence from published literature.

## Methods

This review of systematic reviews followed the Preferred Reporting Items for Systematic Reviews and Meta-Analyses (PRISMA) guidelines for reporting [32]. The study has been registered on PROSPERO (Reg. number: CRD42022371955).

### Search strategy

Our search strategy was designed based on two main concepts: (1) Health System and (2) Fragile and Conflict-affected States (S1 Annex). We conducted searches in Ovid (MEDLINE, Embase, Global Health), Scopus, Web of Science, and the Cochrane Library databases by the

**Table 1. Eligibility criteria.**

| | Inclusion | Exclusion |
|---|---|---|
| **Context** | Fragile and/or conflict-affected states (*ongoing conflict/war or post-conflict*), situations, and their variants/synonyms, also other humanitarian contexts including natural and man-made disasters, and displaced populations | No consideration of fragile and/or conflict-affected states, situations, or their variants/synonyms |
| **Language** | English | Non-English |
| **Availability** | Studies available full text | Abstracts without full text |
| **Type of studies** | Reviews that are conducted systematically (published and pre-prints) | Narrative/traditional/unstructured reviews, observational and experimental studies, study protocols, book chapters, and others. |
| **Subject of the studies** | • Health system strengthening and/or the sub-systems, including the foundations, processes of care and quality impacts addressing wider health system contexts.<br>• Initiatives that contribute to strengthening or supporting health system or delivery of healthcare and is/could be continuous or integrate with the wider system.<br>• Organisation of health system in general, and its sub-systems or the domains and components of health system.<br>• Evaluation of health system strengthening initiatives/interventions. | Temporary public/community healthcare services, response to temporary emergency healthcare needs and, management of individual diseases/conditions, specific healthcare interventions, interventions that are not connected to the wider health system and focused on specific populations groups only, and epidemiological studies. |

final date November 9th, 2022, aiming to retrieve all published literature worldwide in this subject area. There were no restrictions on publication years; however, we only considered literature published in English (Table 1).

## Screening and management

We utilised Rayyan (https://www.rayyan.ai), an online systematic review platform, to facilitate the screening. One reviewer (BB) conducted the initial title and abstract screening. Subsequently, two reviewers (BB and ANMK) independently conducted a duplicate full-text screening of the initially identified articles, applying the inclusion and exclusion criteria. They also independently assessed methodological quality of the selected literature for inclusion in the synthesis using the Joanna Briggs Institute (JBI) Critical Appraisal Checklist for Systematic Reviews and Research Syntheses [33]. Any discrepancies were resolved through discussion between the two reviewers, and when necessary, the rest of the research team (JG and SS) provided input to reach a consensus. The data extracted from the final manuscripts included the author, publication date, context/country profile (e.g., fragile, conflict-affected, other humanitarian crises), name of the included countries, study type, aim/objectives, methods, key findings, conclusions, and limitations. We recorded this information in an Excel spreadsheet. BB extracted the data, and then five randomly selected articles were cross-checked to ensure accuracy; ANMK independently extracted the data for the selected articles and compared it with the data extracted by BB, demonstrating an overall good agreement. Finally, the findings were narratively synthesised using the Lancet's high-quality health system framework (Table 2).

## Results

A total of 2,704 records were initially identified. After removing duplicates, 2005 records were screened for title and abstract, and 53 papers were selected for full-text screening. Twenty-seven systematic reviews met the inclusion criteria and included for synthesis (Fig 2). Hand-searching of the reference lists did not identify any additional relevant paper. The publication years of the included systematic reviews ranged from 2014 to 2022. All papers employed qualitative syntheses. The majority demonstrated high (n = 11) or moderate (n = 14)

**Table 2. Domains, components, and descriptions of high-quality health system framework.**

| Domains | Components | Descriptions |
|---|---|---|
| **Quality impacts** | Better health | Level and distribution of patient-reported outcomes: function, symptoms, pain, wellbeing, quality of life, and avoiding serious health-related suffering. |
| | Confidence in system | Satisfaction, recommendation, trust, and care uptake and retention. |
| | Economic benefit | Ability to work or attend school, economic growth, reduction in health system waste, and financial risk protection. |
| **Processes of care** | Competent care and systems | Evidence-based, effective care: systematic assessment, correct diagnosis, appropriate treatment, counselling, and referral; capable systems: safety, prevention and detection, continuity and integration, timely action, and population health management. |
| | Positive user experience | Respect: dignity, privacy, non-discrimination, autonomy, confidentiality, and clear communication; user focus: choice of provider, short wait times, patient voice and values, affordability, and ease of use. |
| **Foundations** | Population | Individuals, families, and communities as citizens, producers of better health outcomes, and system users: health needs, knowledge, health literacy, preferences, and cultural norms. |
| | Governance | Leadership: political commitment, change management; policies: regulations, standards, norms, and policies for the public and private sector, institutions for accountability, supportive behavioural architecture, and public health functions; financing: funding, fund pooling, insurance and purchasing, provider contracting and payment; learning and improvement: institutions for evaluation, measurement, and improvement, learning communities, and trustworthy data; intersectoral: roads, transport, water and sanitation, electric grid, and higher education. |
| | Platforms | Assets: number and distribution of facilities, public and private mix, service mix, and geographic access to facilities; care organisation: roles and organisation of community care, primary care, secondary and tertiary care, and engagement of private providers; connective systems: emergency medical services, referral systems, and facility community outreach. |
| | Workforce | Health workers, laboratory workers, planners, managers: number and distribution, skills and skill mix, training in ethics and people-centred care, supportive environment, education, teamwork, and retention. |
| | Tools | Hardware: equipment, supplies, medicines, and information systems; software: culture of quality, use of data, supervision, and feedback. |

**Adapted from:** Kruk ME, Gage AD, Arsenault C, Jordan K, Leslie HH, Roder-DeWan S, et al. High-quality health systems in the Sustainable Development Goals era: time for a revolution. Lancet Glob Health. 2018;6(11): e1196-e252.

methodological quality scoring 'yes' to more than or equal to half of the applicable questions in the JBI Critical Appraisal checklist. More than half of them scored 'no' or 'unclear' answers to three questions related to appropriateness of the search strategy (n = 15), criteria for appraising studies (n = 15) and methods to minimise errors in data extraction (n = 14). None of the systematic reviews clearly stated if their critical appraisal was conducted by two or more reviewers independently (S4 Annex).

From the systematic reviews included in our study, 635 source publications in 19 systematic reviews were identified. Across these, 36 sources were duplicated, with one paper included in four reviews, two in three reviews, and the remainder shared between two reviews. Most of the reviews discussed overarching elements of health systems with the main subject of study being healthcare platforms (n = 9), workforce (n = 6), governance (n = 4), coordination (n = 3), community engagement (n = 2), quality of care (n = 2) and general health system reforms (n = 1). Twelve systematic reviews studied conflict-affected contexts, while the rest (n = 15) studied wider contexts, encompassing conflicts, epidemics, pandemics, natural disasters, and other disasters, collectively described as 'humanitarian' or 'fragile and 'conflict-affected'. Six out of those in humanitarian contexts were limited to LMICs.

In this review, about one hundred countries were mentioned, with Democratic Republic of Congo (n = 16), Afghanistan (n = 16) and South Sudan (n = 14) being the most frequent. Fifty-five of the identified countries have been included in the list of World Bank's fragile and conflict-affected situations (2006–2023) at least once. All the sixteen countries (Afghanistan, Burundi, Central African Republic, Chad, Comoros, Democratic republic of Congo, Eritrea, Guinea-Bissau, Haiti, Kosovo, Liberia, Myanmar, Solomon Islands, Somalia, Sudan, and

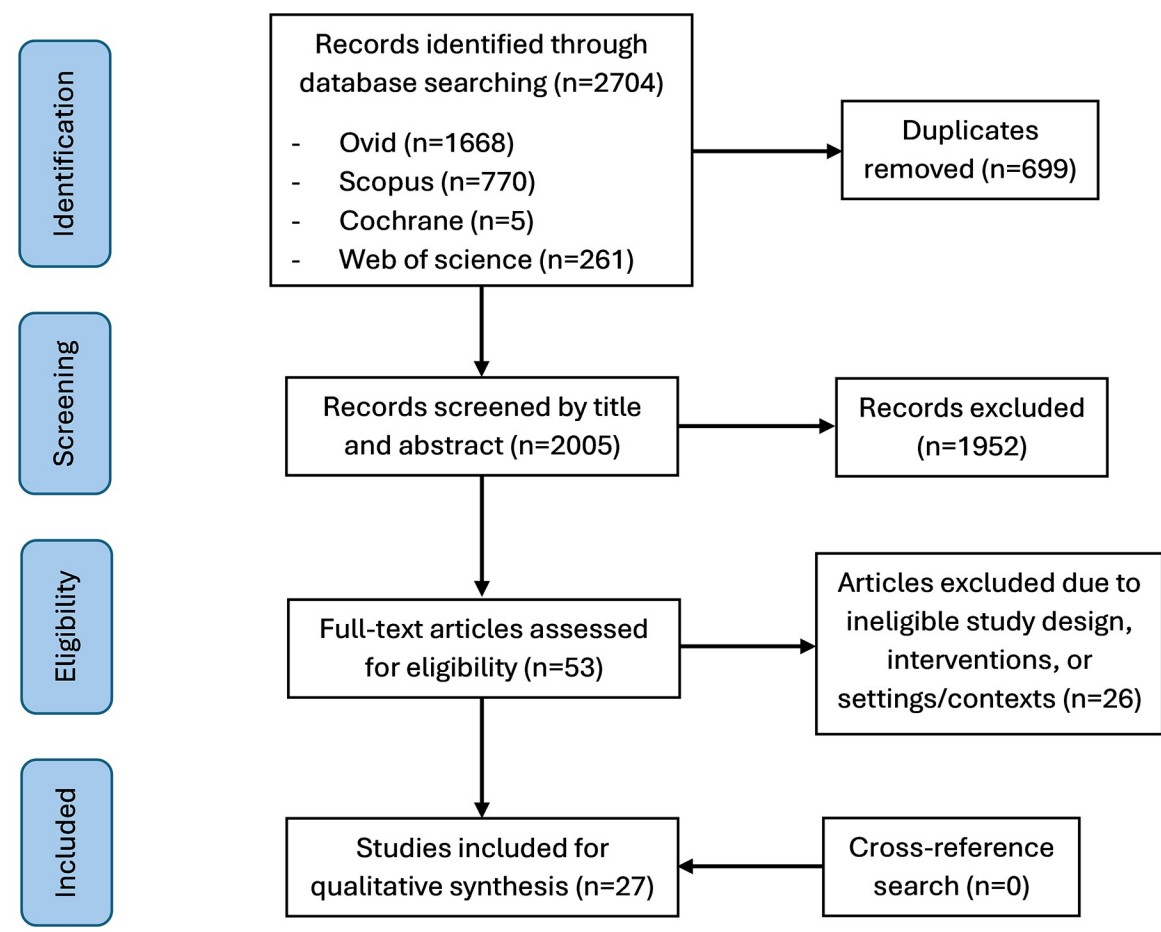

**Fig 2. PRISMA flow diagram [32].**

Zimbabwe) that have consistently been included in the list since its inception were identified in the systematic reviews. The systematic reviews utilised different criteria and definitions of FCAS, of which only four employed the World Bank's list of fragile and conflict-affected situations, and one used triangulation of data from different humanitarian and fragile states classifications including the World Bank's list and fragile states index. Table 3 presents characteristics of the papers with the aims/objectives and relevant summaries of the findings.

The findings of this review are summarised and presented using the domains and components of the Lancet's high-quality health system framework starting with the 'foundations' domain which was evidenced in all the included papers to varying degrees [30].

## I. Foundations

**1. Population.** Twelve systematic reviews explored the significance of cultural norms, community engagement, and collaboration with various community groups in overcoming barriers, improving health outcomes, supporting healthcare providers, enhancing access to healthcare, and promoting health in FCAS. The evidence concentrates in conflict-affected and humanitarian settings predominantly concerning displaced populations and post-conflict contexts in LMICs.

**Table 3. Characteristics of included studies.**

| Study Id. [Ref no.] | Title | Review-ed studies | Shared literature (number) | Contexts | Aim/objectives | Key findings/ HSS initiatives |
|---|---|---|---|---|---|---|
| Jordan et al., 2021 [34] | Quality in crisis: a systematic review of the quality of health systems in humanitarian settings | 55 | Durrance et al. (4) Lokot et al. (3) Asgary et al. (1) Werner et al. (1) Casey (1) | Humanitarian settings in LMICs | To examine the evidence on the quality of health systems in humanitarian settings. | A large gap in the measurement of quality both at the point of care and health system level have been highlighted. |
| Durrance-Bagale et al., 2020 [35] | Lessons from humanitarian clusters to strengthen health system responses to mass displacement in low and middle-income countries: A scoping review | 186 | None | Humanitarian settings/ mass displacement in LMICs | To summarise the scope of the literature on humanitarian cluster interventions responding to mass displacement; and identify lessons from cluster interventions of use in strengthening health system responses to mass displacement. | Non-health humanitarian clusters can contribute to improving health outcomes using focussed interventions for implementation by government or humanitarian partners to inform LMICs health system responses to mass displacement. |
| Lassi et al., 2015 [36] | Impact of service provision platforms on maternal and newborn health in conflict areas and their acceptability in Pakistan: A systematic review | 10 | Roome et al. (1) Casey (2) Ruby et al. (1) | Conflict-affected populations | To undertake a systematic review of global and local (Pakistan) information from conflict areas on platforms of health service provision implemented at community and/or facility level to improve maternal and newborn health within the last ten years (September 2003 to September 2013). | Important steps that can be undertaken to mitigate the effects of conflict on maternal and newborn healthcare services include improvement of provision and access to infrastructure and equipment; development and training of healthcare providers; and advocacy at different levels for free access to healthcare services and introduction of the programme model in existing health system. |
| Asgary et al., 2022 [37] | A systematic review of effective strategies for chronic disease management in humanitarian settings; opportunities and challenges | 48 | Bowsher et al. (4) Jordan et al. (1) Ruby et al. (2) | Various humanitarian settings | To systematically assess the evidence for interventions and identify effective strategies for management of NCDs and ultimately guide the related research, programme planning, and policies in the humanitarian settings. | Models of care were largely not well-described and varied between studies due to contextual constraints. |
| Vivalya et al., 2022 [38] | Developing mental health services during and in the aftermath of the Ebola virus disease outbreak in armed conflict settings: a scoping review | 7 | None | Ebola Virus Disease outbreak in a conflict-affected settings | To design the basis of improving mental health services via the integration of mental health into primary healthcare in the Democratic Republic of Congo. | The need of involving mental health specialist and non-specialist (including the community) in treating patients with mental health problems has been highlighted. |
| Homer et al., 2022 [39] | Enhancing quality midwifery care in humanitarian and fragile settings: A systematic review of interventions, support systems and enabling environments | 24 | Lin et al. (1) Miyake et al. (2) | Unspecified humanitarian and fragile settings | To identify the factors affecting an enabling environment for midwives in humanitarian and fragile settings and to explore the availability and effectiveness of support systems for midwives. | Facilitators to an enabling environment were community involvement and engagement and an adequate salary, incentives, or benefits; and support systems were education, professional development, supportive supervision, mentorship, and workforce planning. |

(*Continued*)

**Table 3.** (Continued)

| Study Id. [Ref no.] | Title | Review-ed studies | Shared literature (number) | Contexts | Aim/objectives | Key findings/ HSS initiatives |
|---|---|---|---|---|---|---|
| Lokot et al., 2022 [40] | Health system governance in settings with conflict-affected populations: a systematic review | 34 | Jordan et al. (3) Roome et al. (1) | Settings with conflict-affected populations including refugees, asylum seekers, internally displaced and conflict-affected non-displaced populations | To examine existing evidence on health system governance in settings with conflict-affected populations globally. | The most common facilitators of governance were collaboration between stakeholders, bottom-up and community-based governance structures, inclusive policies, and longer-term vision. |
| Casey, 2015 [41] | Evaluations of reproductive health programmes in humanitarian settings: a systematic review | 36 | Werner et al. (3) Jordan et al. (1) Lassi et al. (2) Roome et al. (1) Beek et al. (2) | Humanitarian crises in LMICs | To explore the evidence regarding reproductive health services provided in humanitarian settings. | Reproductive health programmes can be implemented in challenging settings, and that women and men will use the services when they are of reasonable quality. |
| Durrance-Bagale et al., 2022 [42] | Community engagement in health systems interventions and research in conflict-affected countries: a scoping review of approaches | 19 | Jordan et al. (4) Abujaber et al. (1) | Conflict-affected settings | To identify and interrogate the literature on community engagement in health system interventions and research in conflict-affected settings. | Engagement of communities in identifying and setting priorities, decision-making, and implementing health system interventions highly contribute to the success of healthcare provision. |
| Miyake et al., 2017 [43] | Community midwifery initiatives in fragile and conflict-affected countries: a scoping review of approaches from recruitment to retention | 23 | Roome et al. (3) Homer et al. (1) *(Included in Lin et al.)* | FCAS based on the World Bank's list of fragile situations | To examine community midwifery approaches, from recruitment to retention, in FCAS. | Community midwifery is beneficial in increasing access for marginalised and hard-to-reach communities and potential value for health equity and sustainable development targets in challenging settings. |
| Ismail et al., 2022 [44] | Strengthening vaccination delivery system resilience in the context of protracted humanitarian crisis: a realist-informed systematic review. | 50 | None | Humanitarian setting in LMICs, defined according to World Bank Country and Lending Groups classification | To critically evaluate evidence on the effectiveness of system-level interventions for improving vaccination coverage in protracted crises, focusing on how they work, and for whom, to better inform preparedness and response for future crises. | Strengthening the resilience of vaccination delivery systems depends on system adaptation across a range of areas, including bolstering access through strengthened outreach, multiple service pathways and better integration with other essential services, as well as demand-generation activities. |
| Rayes et al., 2021 [45] | Policies on return and reintegration of displaced healthcare workers towards rebuilding conflict-affected health systems: a review for The Lancet-AUB Commission on Syria | 13 | Roome et al. (1) | Post-conflict health systems | To identify policies and policy recommendations intended to facilitate the return of displaced healthcare workers to their home countries and acknowledge their contribution to rebuilding the post conflict health system. | No explicit programmes or initiatives focused on policy levers to encourage the return of displaced healthcare workers, and health workforce rebuilding have been identified; however, health system rebuilding, and rehabilitation have been demonstrated to serve as precursors and reinforcers of the successful return, repatriation, and reintegration of displaced healthcare workers. |

*(Continued)*

**Table 3.** (Continued)

| Study Id. [Ref no.] | Title | Review-ed studies | Shared literature (number) | Contexts | Aim/objectives | Key findings/ HSS initiatives |
|---|---|---|---|---|---|---|
| Roome et al., 2014 [46] | Human resource management in post-conflict health systems: review of research and knowledge gaps | 56 | Lassi et al. (1) Lokot et al. (1) Miyake et al. (3) Lin et al. (3) Casey (2) Rayes et al. (1) *(Included in Lin et al.)* | Post-conflict health systems | To present a global review of published research on human resource management in post-conflict health systems during the years, 2003–2013. | Identified initiatives to strengthen post-conflict workforce include: • Expatriate recruitment to fill the workforce gap • Good leadership to ensure the commitment of major actors in strengthening the workforce • Availability of human resource data to inform recruitment and workforce planning decisions • Healthcare workers pay reforms • Rebuilding and upgrading pre-service education and training provision • Intensive and sustained in-service retraining and up-skilling • Formal deployment systems • Direct financial incentives such as pay and bonuses, and indirect financial and non-financial incentives to attract healthcare workers to rural and less secure areas • Job descriptions that are centrally coordinated • Task shifting to mitigate shortages of trained and qualified healthcare workers • Strong and skilled management capacity at all levels as to improving work performance • Performance appraisal • Performance-related financial and non-financial incentives. |
| Bertone et al., 2018 [47] | Context matters (but how and why?) A hypothesis-led literature review of performance-based financing in fragile and conflict-affected health systems | 140 | None | FCAS based on the World Bank's list of fragile situations | To interrogate existing grey and published literature on how the FCAS context influences the adoption, adaption, implementation, and health system effects of PBF in order to support or refute a set of hypotheses about their interaction. | There is some (limited) evidence of health system effects of performance-based financing which may be contextually driven. |

*(Continued)*

**Table 3.** (*Continued*)

| Study Id. [Ref no.] | Title | Review-ed studies | Shared literature (number) | Contexts | Aim/objectives | Key findings/ HSS initiatives |
|---|---|---|---|---|---|---|
| Chol et al., 2018 [48] | Health system reforms in five sub-Saharan African countries that experienced major armed conflicts (wars) during 1990–2015: a literature review | 70 | None | Conflict-affected low-income sub-Saharan African countries | To understand the best health system practices in five SSA countries that experienced wars during 1990–2015, and yet managed to achieve a maternal mortality reduction–equal to or greater than 50% during the same period. | Reforms across all five countries (Angola, Eritrea, Ethiopia, Mozambique, and Rwanda) were mainly driven by political commitment within each country to decentralise health systems and health workforce innovations such as community health workers' programmes. |
| Akl et al., 2015 [49] | Effectiveness of mechanisms and models of coordination between organisations, agencies and bodies providing or financing health services in humanitarian crises: A systematic review | 4 | Lotfi et al. (1) | Various humanitarian crises | To assess how, during and after humanitarian crises, different mechanisms, and models of coordination between organisations, agencies and bodies providing or financing health services compare in terms of access to health services and health outcomes. | It has been suggested that information coordination between organisations, agencies, and bodies providing healthcare services in humanitarian crises settings may be effective in improving health systems inputs; and management and directive coordination such as the cluster model may improve health system inputs, in addition to access to healthcare services. |
| Lotfi et al., 2016 [50] | Coordinating the provision of health services in humanitarian crises: A systematic review of suggested models | 34 | Akl et al. (1) | Various humanitarian crises | To identify published models of coordination between organisations funding or delivering health services in situations of humanitarian crisis worldwide. | Five coordination models (Cluster approach, 4Ws, the Sphere project, the 5×5 model and the model of information coordination) have been implemented worldwide in different disasters in order to coordinate the delivery of healthcare services, and it is challenging to provide specific guidance on which model to use. |
| van Daalen et al., 2022 [51] | Impact of conditional and unconditional cash transfers on health outcomes and use of health services in humanitarian settings: A mixed-methods systematic review | 23 | None | Various humanitarian settings | To evaluate the evidence on the effect of cash transfers on health outcomes and health service utilisation in humanitarian contexts. | There is evidence to suggest a positive relationship between cash transfers and health outcomes in humanitarian settings. |
| Schmid et al., 2022 [52] | Models of care for non-communicable diseases for displaced populations in Iraq: a scoping review | 22 | None | Humanitarian setting/ displaced population in Iraq | To explore models of NCD care for displaced populations in Iraq, in order to build evidence to design context adapted models of care. | There is a scarcity of evidence on the effectiveness of models of NCDs care for displaced populations in Iraq, calling for capacity building initiatives focused on implementation research and evaluation. |
| Werner et al., 2022 [53] | The role of community health worker-based care in post-conflict settings: a systematic review | 55 | Casey (3) Jordan et al. (1) Abujaber et al. (1) Beek et al. (1) | Post-conflict settings | To characterise systematically the literature on the role of community health worker-delivered healthcare in fragile and conflict-affected settings and synthesise reported information on the effect of these interventions on key healthcare functions. | Community health worker interventions are effective and efficient in circumventing the barriers associated with access to care and improve health outcomes. |

(*Continued*)

**Table 3.** (Continued)

| Study Id. [Ref no.] | Title | Review-ed studies | Shared literature (number) | Contexts | Aim/objectives | Key findings/ HSS initiatives |
|---|---|---|---|---|---|---|
| Ruby et al., 2015 [54] | The Effectiveness of Interventions for Non-Communicable Diseases in Humanitarian Crises: A Systematic Review | 8 | Asgary et al. (2) Casey (1) Lassi et al. (1) *(Included in Asgary et al.)* | Various humanitarian crises in LMICs | To systematically review evidence on the effectiveness of interventions targeting NCDs during humanitarian crises in LMICs. | The use of standardised algorithms that can be implemented consistently and monitored via patient tracking using electronic medical records has been shown successful. |
| Lin et al., 2022 [55] | Health care worker retention in post-conflict settings: a systematic literature review | 25 | Roome et al. (3) | Post-conflict settings | To determine the availability of evidence related to the retention of healthcare workers in fragile and post-conflict settings and evaluate what actions might be successful in improving retention and reducing attrition. | Adopting policies that focus on improving financial incentives, providing professional development opportunities, establishing flexibility, and identifying staff with strong community links may ameliorate workforce attrition. |
| Dobiesz et al., 2022 [56] | Maintaining health professional education during war: A scoping review | 56 | None | Conflict-affected/ active war | To describe the scope of barriers and targeted interventions to maintaining health professional education during war and summarise the research. | Identified interventions, but not evaluated for their success were: (1) Curriculum: shortening of training time, and the use of telemedicine and web-based educational platforms (2) Resources: moving to less dangerous alternative locations for instruction, and the use of online resources (3) Personnel: intervention efforts to recruit both students and faculty and to maintain the academic calibre of personnel (4) Wellness: wellness initiatives and working with behavioural health specialists; and transportation and safe passage to classes and hospital sites (5) Oversight: development of enhanced national standards and organising medical bodies to ensure maintenance of adequate and uniform educational standards. |
| Beek et al., 2017 [57] | A review of factors affecting the transfer of sexual and reproductive health training into practice in low and lower-middle income country humanitarian settings | 7 | Casey (2) Werner et al. (1) | Humanitarian settings in LMICs | To review evidence from the literature to determine which factors influence the transfer of training on sexual and reproductive health in humanitarian settings. | Factors to facilitate the transfer of learning include appropriate resourcing, organisational support, and confidence in healthcare workers, and these are present at the individual, training, organisational, socio-cultural, political and health system levels. |

*(Continued)*

**Table 3.** (Continued)

| Study Id. [Ref no.] | Title | Reviewed studies | Shared literature (number) | Contexts | Aim/objectives | Key findings/ HSS initiatives |
|---|---|---|---|---|---|---|
| Abujaber et al., 2022 [58] | Examining the evidence for best practice guidelines in supportive supervision of lay health care providers in humanitarian emergencies: A systematic scoping review | 11 | Durrance et al. (1) Werner et al. (1) | Humanitarian emergencies | To identify empirically supported features of supervisory practices for lay health care providers in humanitarian emergencies, towards a stronger evidential basis for best practice in supportive supervision. | The use of supportive supervision for improved client clinical outcomes, healthcare service sustainability, lay healthcare worker well-being and performance have been highlighted. |
| Bowsher et al., 2021 [59] | eHealth for service delivery in conflict: a narrative review of the application of eHealth technologies in contemporary conflict settings | 46 | Asgary et al. (4) Winders et al. (1) | Settings of active conflict or ongoing insurgency | To examine the current state of usage of eHealth technologies on healthcare delivery in contemporary conflict settings in order to identify categories of usage and highlight evidence gaps in the application of eHealth in these contexts. | The use of eHealth presents a clear scope for the delivery of medical care, and the role of technological innovation in addressing escalating health challenges. |
| Winders et al., 2021 [60] | The effects of mobile health on emergency care in low- and middle-income countries: A systematic review and narrative synthesis | 46 | Bowsher et al. (1) | Humanitarian settings in LMICs | To provide the first formal synthesis of the effects of mHealth as it relates to emergency care in LMICs. | mHealth is effective in improving provider-focused, process-driven, and patient-centred outcomes in both routine and complex emergency care settings in LMICs through ensuring the delivery of consistent and high-quality care while improving access to healthcare services. |

The cultural norms of patients have played a role in shaping healthcare services and over-coming cultural and religious barriers is needed to ensure effective care [34–37]. Engaging the community in HSS initiatives, particularly in post-disaster recovery has been instrumental. Enhancing health outcomes, in relation to mental health, as exemplified by armed conflict settings suffering from the Ebola virus disease outbreak and its aftermath, required involving relatives with community health workers in closely monitoring individuals experiencing psychological distress [38]. Engaging refugee healthcare workers and recruiting them in facilities serving refugee and displaced populations was also crucial in supporting the workforce in post-conflict humanitarian settings [40, 41]. Training the community, particularly displaced populations such as refugees and asylum-seekers, was found to address healthcare gaps, and promote health by training others and distributing health information [35, 37, 40, 41]. Health promotion activities organised and provided through community groups, including churches, also reduced stigma and created a more welcoming atmosphere in healthcare facilities [42]. In addition, involving the community in selecting candidates for community health workers training increased their acceptance and facilitated their work on deployment [43].

Community cohesion and pooled resources played a crucial role in mitigating the impacts of crises [34]. Collaboration between the community and health system was also essential in providing a safe environment for healthcare workers [39]. Communities also have provided financial support in situations where the government was unable to pay healthcare workers' salaries [39] in addition to facilitating access to healthcare services by organising patient transportation [42]. Involvement of community volunteers, and intensive community engagement activities improved trust and uptake of healthcare interventions such as immunisation in

protracted crises settings [44]. Community-driven multistakeholder programmes were found to be effective in facilitating increased access to healthcare services and improving community healthcare initiatives. Community-nominated volunteers played a significant role in designing specific healthcare services and health promotion activities. Hence, involving traditional community leaders and empowering the community as service users to propose their own solutions was emphasised as a means to enhance healthcare delivery, ensure cultural appropriateness, and overcome resistance [42]. Faith-based organisations, including churches, also played an influential role in reaching out to migrant healthcare workers and mobilising them to return to their home countries to strengthen the post-conflict health systems in their respective countries [45]. Furthermore, communities have played a role in mobilising political will for quality healthcare services [34].

Overall, the evidence suggests that it is important to address cultural and religious barriers; and foster collaboration and support as to promote culturally appropriate healthcare and enhance community engagement through individuals and organised community groups. Empowering, and involving refugees and displaced populations in the health system also contribute to promoting health, narrowing the health workforce gaps and improve healthcare utilisation. This suggests the importance of coproduction and proactive engagement of the community in strengthening the health system.

**2. Governance.**   Twelve systematic reviews highlighted the challenges in health system governance and coordination in various FCAS including acute and long-standing humanitarian disaster and post-disaster settings, as well as in countries rebuilding their health system long after armed conflict. They explored various barriers, and facilitators of health system governance; and aspects of leadership and financing were most evidenced.

The study by Lokot et al. [40] identified the challenges in health system governance and coordination in settings with refugees, displaced and non-displaced conflict-affected populations. These challenges included unclear responsibilities of different actors, mistrust among stakeholders and within vertical programme delivery, tension, and lack of harmonised health response; as well as lack of and restrictive healthcare policies, implementation problems, lack of expertise and capacity, and limited legal guidance. Financial barriers were frequently cited in the reviews. They included fragmented structures such as the lack of inclusive finance systems, inefficient resource allocation, and a lack of clarity on financial needs [40]. Additionally, having a disorganised structure of service delivery and limited infrastructure, including health facilities, water supply and other utilities were also identified as challenges in various fragile and humanitarian settings [39]. Government agencies were not trusted due to perceived lack of accountability, unethical practices such as reporting non-existent needs to obtain funding and prioritising cost-effectiveness over meeting medical needs; transparency-related barriers, such as employment issues, were also identified [40]. Post-conflict, weak governance and financial support were reported to influence decisions regarding transfer or deployment of human resources. Existing job descriptions became irrelevant, leading to the drafting of new job descriptions by NGOs and aid agencies to meet immediate service needs which made it difficult to manage performance of healthcare workers and the increasing number of job descriptions without central coordination [46]. In settings with conflict-affected populations, multiple humanitarian actors existed, although poorly coordinated, with fragmented and non-uniform coordinating hubs, and third parties conducting monitoring and evaluation [40]. There was evidence that NGOs fill government gaps; however, donors were found to be central to both the problems and solutions as their interests may dominate funding, decision-making, and policymaking [40, 47].

Governance improvements, particularly political commitment and enhanced leadership were reported to improve health system performance in various fragile and conflict-affected

settings. This included people-centred governance approaches which improved stakeholder engagement, accountability, setting a shared strategic direction, and stewarding resources responsibly [34, 39, 40, 42]. Additionally, there was evidence that decentralisation supported health system reforms in conflict-affected contexts [40, 48]. Management and directive coordination, such as the United Nations Humanitarian Cluster Approach, 4Ws, the Sphere project, the 5×5 model and the model of information coordination improved coordination amongst humanitarian healthcare organisations; however, providing specific guidance on which model to use was found to be challenging [40, 49, 50]. The cluster system contributed to improving access to healthcare services and other inputs, such as drug availability, workforce, and the level of healthcare services by improving the coordination amongst different health organisations particularly in acute humanitarian crisis [49]. The non-health humanitarian clusters also contributed to improving health outcomes in displaced populations [35]. Additionally, prior networks and infrastructure built through previous work, including governance structures and outreach capabilities were reported to be crucial to strengthening community health service programmes such as vaccination, especially in less secure areas. Cross-border governance and coordination initiatives among different countries governments have also been implemented to reduce the risk of vaccine-preventable diseases transmissions [44]. In conflict-affected settings, structural reforms in adopting service delivery, using information from various sources and databases for sharing and policymaking; as well as having prior systems and plans in place to manage the changing needs and emergency preparedness with strong support from the central level facilitated resource deployment at the local level [40].

Health financing strategies such as community-based health insurance, performance-based financing, coordinated donor funding, and fees retention policy for local health facilities assisted with health system reforms and services in an ongoing and near post-disaster situations, as well as long-term health system reforms [44, 48]. 'Cash transfers', that is forms of payment by formal or informal institutions to people, were also applied by governments and humanitarian agencies/NGOs with the purpose of reducing economic hardships and poverty, increasing households' access to basic food, improving food security, or preventing acute malnutrition [51]. Hence, creating a reliable and cost-effective financial monitoring system, together with better coordination within and across organisations, and donors, was recommended to strengthen the system of managing the outpouring resources of aid in various humanitarian settings [50]. 'Performance-based financing', which means payments and incentives provided directly to healthcare workers or facilities, had a positive effect on healthcare interventions [44]. Performance-based financing has been more common in FCAS contexts and more commonly adopted earlier in the recovery period [34, 47]. It has been shown that, conditions of fragility may favour its rapid emergence; and the key factors that drove the adoption included the strong influence of external actors; greater openness to institutional reforms; lack or lower level of trust within the public system and between government and donors; widespread corruption; lack of accountability and fiduciary concerns; as well as the perceived lack of capacity of the government to manage external funds which all tend to favour more contractual approaches [47]. Although the situation of de facto decentralisation also drove the adoption of performance-based financing, the lack of formal or appropriate decentralisation processes has been shown to have hindered progress as external influence solely was not enough to sustain the project. Therefore, it has been suggested that performance-based financing may be more likely to be sustained in countries where it is introduced as part of a wider set of health system and health financing reforms. Hence, it has been recommended as a tool to reinforce the overall governance of a health system [47]. Furthermore, evidence suggests the importance of countries owning their health finance, in order to minimise the negative influences of external donors on the provision of healthcare and strengthen the health system in the long run [48].

In summary, the evidence suggests that health system governance in FCAS faces significant challenges including unclear responsibilities, mistrust, financial barriers, and inadequate infrastructure. It underscores the importance of effective governance improvements, political commitment and leadership to enhance health system performance. These suggest the need for comprehensive and people-centric governance strategies, effective coordination mechanisms, and appropriate financing models. Additionally, the findings emphasise the significance of country ownership in health financing to minimise external influences and strengthen overall health system governance for sustained success.

**3. Platforms.** Ten systematic reviews highlighted the challenges and barriers of access and healthcare service platforms in crises situations, and the impact on vulnerable populations. Various care models and strategies to strengthen the health service platforms in different context of FCAS were reviewed.

Providing adequate healthcare services in crises settings presented several challenges due to contextual constraints and feasibility opportunities; and higher rates of healthcare service interruption were observed immediately post-disaster [37, 41]. Active conflicts and concerns with security and safety of patients and healthcare workers had strong influence on the choice of health service platforms [44]. They also became significant barriers to healthcare provision; for instance, community-based services and night-time work became overly unsafe [39]. Apart from this, multiple factors hindered access to healthcare, particularly affecting vulnerable populations such as people from minority groups. These factors included the nature and burden of specific diseases, especially NCDs, as well as non-functioning facilities, inadequate staffing, distance to the nearest healthcare facility, and an influx of refugees or returnees, also recently, the COVID-19 lockdowns [37, 52]. Limited access to healthcare has commonly been noted during active conflicts, influxes of displaced populations, and pandemic related restrictions. Besides, availability of medicines is a crucial aspect of access to healthcare, with public facilities generally performing poorer compared with camp-based primary healthcare services or private healthcare providers in settings with displaced populations [52]. The findings also highlighted significant variations in access rates across different areas and times, even within the same country, supporting the suggestion on the need of understanding local circumstances when designing healthcare models [38, 52]. Additionally, several barriers, including financial, logistical, organisational, and sociocultural factors, as well as the context of security, hindered implementation or success of new interventions [37]. The stage of humanitarian crisis was also found to be a critical factor in determining the feasibility of different healthcare models as for instance, disaster situations, particularly ongoing or recent conflicts impact people's healthcare seeking behaviour [52].

Significant variations of healthcare models were found in humanitarian settings. Some NCD interventions were integrated into primary healthcare systems, while others were implemented as vertical or single interventions [37]. The most frequently described model in settings with displaced populations in Iraq involved a joint response by camp-based or parallel structures and the formal national health system [52]. Ruby et al. [54] documented the utilisation of traditional treatments for arthritis in refugee settings, noting improved health outcomes when compared to Western medicine. Establishing a strong relationship between modern mental healthcare and traditional or religious healers have also been found to be beneficial [38]. Furthermore, coordination with traditional healthcare providers, and even recruiting them into the health system supporting regular healthcare providers has reported to improve access to healthcare in humanitarian settings [39].

Strategies to strengthen vaccination delivery in protracted humanitarian settings commonly included utilising multiple service delivery pathways such as fixed-sites, mobile teams, and mass-vaccination sites. Central coordination led by the Ministry of Health, in collaboration

with agencies and NGOs, was found to facilitate on-the-ground preparedness and delivery of immunisations before disease outbreak [44]. Lassi et al. [36] included the use of community-based maternal and neonatal health services, which involved training community health workers, conducting awareness workshops, and empowering the community through communication and education. Accordingly, outreach services, involving trained physicians and community health workers, were employed to raise community awareness, transfer skills to the district health team, overcome religious and cultural barriers, and motivate health facilities in conflict-affected areas [36]. Community health worker interventions have been reported to improve access to healthcare, notably in psychosocial support, and reproductive and maternal health services, post-conflict [53]. Building the capacity of local community health workers and involving them in health promotion initiatives were reported as key in ensuring the continuity of essential healthcare services [42]. Trained mobile healthcare workers also provided basic emergency obstetric and new-born care along with a full range of family planning services in humanitarian settings [41]. According to Ismail et al. [44], the use of mobile health teams improved vaccination coverage in humanitarian crisis. They also emphasised the importance of strengthening public-private partnerships by integrating private providers into routine delivery of immunisation. However, Asgary et al. [37] argued that, establishing new clinics, training healthcare workers, and supporting stationary clinics are more effective in addressing NCDs than relying solely on mobile clinics. Yet, mobile clinics established immediately after disasters were more inclined to address acute injuries, whereas pre-existing stationary clinics were more likely to assist patients seeking NCDs care in the same timeframe [37].

In summary, the evidence suggests that healthcare service platforms face a number of challenges, mainly due to the context and feasibility of interventions in providing adequate healthcare services to vulnerable populations, and access to healthcare being constrained in FCAS. Strengthening healthcare infrastructure, improving access to medicines and supplies, capacity building, and integrating private and traditional healthcare providers in the system can enhance healthcare service delivery. Furthermore, it is crucial to focus on tailored and context-specific healthcare models that consider local circumstances and address specific challenges while planning interventions.

**4. Workforce.** Thirteen systematic reviews considered barriers, facilitators, and recommendations to strengthen the health system workforce in FCAS. The challenges of attracting and retaining healthcare workers, and health professional training and education, as well as workforce strengthening in post-conflict health systems were most evidenced.

Conflict-affected settings faced unique challenges in terms of health workforce. During conflicts, national health systems and healthcare workers often become targets, leading to displacement, injuries, or even death because of targeted attacks or collateral damages [39, 45, 46, 55, 56]. In such settings, healthcare workers tend to leave their jobs or country, seeking safety and recognition for their skills elsewhere. Security concerns greatly impact staff retention and job satisfaction, with personal safety being a key factor in the decision to stay or leave [39]. The fear of instability and political challenges also push many to decide to leave their job or their home country and refuse to be deployed in certain areas [55]. Consequently, healthcare workers become concentrated in safer regions, leaving remote and conflict-affected areas underserved [46]. External factors, such as better opportunities and technologies in host countries, also influence migration decisions [46, 55]. Post-conflict, the flight of healthcare workers may persist due to skill mismatches and disrupted healthcare services [46]. Uncertainty about the situation and the potential re-emergence of conflict was found to be strongly associated with healthcare workers decision to migrate or not; also discouraged them from returning to their home countries. Destruction of health facilities and staff residences further complicated redeployment [43, 46, 55]. Intentions to migrate were also driven by challenging work

environments, inadequate resources, overwhelming responsibilities, and unmanageable work-loads [55]. Furthermore, security concerns posed significant barriers to healthcare provision; community-based services and night-time work became particularly unsafe. Although safety concerns related to security included both physical and psychological issues, the literature mostly focused on the physical aspect [39].

Salaries, incentives, and benefits also played crucial roles in the supply and retention of healthcare workers [39, 46, 55]. It has been observed that, inadequate salaries often force healthcare workers to hold multiple jobs, which inadvertently affects the quality of healthcare they provide [39, 46]. However, governments were typically unable to offer competitive and timely salaries; and salary scales were constrained by the amount of donor financing available [39, 44, 46]. Consequently, NGOs and aid organisations unintentionally contributed to labour market imbalances by attracting public healthcare workers to work for them, with the offer of more attractive employment packages, potentially concentrating the workforce in safer geo-graphic areas depriving the relatively unsafe areas, also reducing the size of workforce available to restore routine public healthcare services [46, 55]. Post-disaster, many NGOs cease to func-tion in full capacity and there may only be limited workforce vacancies due to the weakened economy and health system. The remaining NGOs can also not afford or sustain their work-force, possibly leading to further labour market disequilibrium [55]. Additionally, the growing market for private healthcare in post-conflict settings may challenge the dominance of the public health sector [46].

Despite the challenges, available evidence suggests that some healthcare workers chose to remain in conflict-affected settings or return from migration due to personal connections, community support, intrinsic motivations, and favourable working conditions [45, 55]. At times, policymakers have introduced programmes to reintegrate returned healthcare workers; however, it appears that, logistical, and financial barriers, legal status, and certification posed significant challenges to their reintegration [45]. In some areas, the returning healthcare work-ers had a better chance of employment due to the better education they may have gained; however, there is evidence that feelings of emotional resentment from, or even being threat-ened by those who did not leave; and working through this situation had been challenging [45]. Moreover, their return contributed to an oversaturated health labour market encouraging students and more recent graduates to migrate abroad, and furtherly impact the remaining healthcare workers [45, 55, 56]. On the other hand, expatriates have been recruited to provide short-term solutions [46, 48]. However, their recruitment resulted in high turnover, discrepan-cies in skills and qualifications, resentment over control and salary differences, and limited opportunities for local healthcare workers. Additionally, the expatriates also failed to transfer skills to the national staff, and potentially hinder those willing to return to their countries in the long run [46].

A lack of training and education opportunities has been reported as a common push factor for healthcare workers, and a number of other factors impacted education and training [45, 55]. Conflict and war posed significant challenges in providing and maintaining education for healthcare professionals because of disruptions in access to educational resources and infrastructure, as well as threats to the physical and psychological well-being of staff and stu-dents [56, 57]. Teaching institutions also faced limitations in their training capacities due to the loss of teachers and a lack of skilled trainers [39, 43, 55, 56]. The quality of ad hoc or emer-gency on-the-job training provided by NGOs and local providers to fill the gap has also been criticised for its poor quality and limited impact [40, 46]. In response to these challenges, some governments have sent individuals to other countries for training, with the expectation that, they will return to serve in the national public health sector [45]. Deployment of locally trained new graduates to rural and underserved areas as a post-graduation requirement was also

implemented in conflict-affected sub-Saharan African countries [48]. During times of war, interventions, such as the reduction of training duration, have been implemented. Many countries have also adjusted their curricula to align with the situation, incorporating the management of war casualties, trauma care, infectious diseases, and mental health. Additionally, higher-level training programmes have been adapted to meet military requirements, specifically addressing areas such as surgery and anaesthesia care [56].

Moreover, task shifting and training community health workers in general or specific areas of healthcare, have been implemented as strategies to address shortages of trained and qualified healthcare workers in various fragile contexts [43, 46, 48, 53, 55]. However, cultural norms, social situations and security concerns have hindered recruitment and deployment/transfer of the newly qualified healthcare workers in some geographic areas, and female healthcare workers, in particular, faced additional challenges; they were discouraged from working in unstable areas, as they faced higher risks [46, 58]. Hence, retaining female healthcare workers has been difficult, mainly in the least-developed areas with limited road connections [43, 55]. Furthermore, some conservative communities also did not believe that young unmarried women should become healthcare workers [39]. Besides, women and girls had limited access to education, especially in rural areas, and candidate eligibility to community health workers training has been problematic [43]. Yet females are shown to play a vital role; in some conflict-affected areas, only the presence of a female community health workers was associated with increased skilled birth attendance and higher family planning use [41] and female healthcare workers tend to remain in their local areas and continue to work through conflict more frequently [55].

In relation to standardising training and education programmes, a lack of consensus among the national stakeholders regarding their value and future posed a significant obstacle, which was often underestimated by governments [55, 56]. The absence of clear definitions of the roles and responsibilities among the key governmental stakeholders also impeded the healthcare workers education and training [43]. Curriculum development, and the selection of students and faculty were subjected to interference from the governmental bodies, the military, and medical organisations [43, 56]. Furthermore, governments occasionally redirected funds allocated for education and public health initiatives towards military funding or even prohibited education and research initiatives altogether. Inadequate educational accreditation, coupled with the absence of a regulatory system and licensing examination, hindered the assessment of graduates' competency [43]. In conflict-affected settings, training standards and quality assurance were compromised due to the unregulated privatisation of training providers, resulting in an oversupply of graduates who anticipated employment in the public sector [46]. The absence of national standards and quality assurance in education and training institutions had serious implications for the quality of healthcare workers produced. The recommendations to address these issues included enhancing national standards and organising medical bodies to ensure the maintenance of adequate and uniform standards [56]. Additionally, it was proposed to establish educational accreditation linked to clear performance objectives [43] and to establish strategic partnerships with international institutions to provide relevant opportunities for accreditation [55].

In summary, FCAS, particularly in the context of conflict suffer from challenges such safety and security, living and working conditions, limited education and training, and education quality assurance and standardisation of healthcare workers. Prioritising the safety and security of healthcare workers, providing competitive salaries and incentives, equitable distribution of healthcare workers, and investment in training and education programmes are considered important to attract, motivate, and retain healthcare workers. Rebuilding healthcare infrastructure, improving working conditions, and ensuring access to resources help retaining

healthcare workers and support the quality of healthcare services. Additionally, standardisation, accreditation, and strategic partnerships can help ensuring the quality and effectiveness of health professional education and training that produce competent professionals.

**5. Tools.**   Eleven systematic reviews discussed both the hardware and software aspects of health system tools focusing on the gaps in physical resources and supervision, as well as interventions and eHealth tools to address the challenges in various contexts of FCAS.

The study conducted by Jordan et al. [34] highlighted physical resources necessary for a functioning health system, such as medicines and other supplies, often served as healthcare quality indicators and they were lacking in humanitarian settings. Inadequacy of equipment, supplies, and infrastructure in clinical settings and training institutions, coupled with insufficient supportive supervision adversely affected the quality of health professional education in conflict-affected settings, and contributed to attrition of healthcare workers [46, 55, 56, 58]. Abujaber et al. [58] reported that supportive supervision played a crucial role in identifying areas requiring development and capacity building in humanitarian emergencies. This led to improved service delivery and performance, particularly among lay healthcare workers. However, healthcare facilities, particularly in rural areas, frequently lacked strict monitoring and supervision systems impacting the services such as by delaying the delivery of materials [43]. Significant gaps in supervision also included supervision by non-clinical personnel, and irregularity or absence of supervision, mainly due to problems with security. Often in FCAS, supervision was also regarded as a means to collect medical records data and provide administrative feedback, rather than offering support to healthcare workers [39]. Besides, the number of capable managers and supervisors was severely limited in crisis situations [43, 46]. Nevertheless, remote supervision was found to yield similar results as in-person supervision [58]. According to Roome et al. [46], these issues were seldom prioritised in long-term training and capacity building strategies, and opportunities to enhance management skills during post-conflict reconstruction were overlooked. Consequently, untrained, under-resourced, and unsupported managers were entrusted with addressing workforce problems while simultaneously coping with the aftermath of conflict. Hence, implementing policies and interventions, such as short-term training, practical tools in a short course format, didactic training, on-site projects, mentoring, and securing high-level support has been suggested to ensure active participation [46].

Conflict-affected areas were characterised by poor-quality data, and inconsistent data collection [40]. Inefficient or inexperienced bodies contributed to incomplete, flawed, or manipulated human resources data, further delaying health system reforms [46]. However, eHealth tools, such as mobile applications were used as interventions for information management, including health records and data in active conflict settings [59]. Information and communication technologies also played a supportive role in health and medical care during disaster responses [49] also aided cash transfer programmes [60]. Compared to traditional paper-based methods, mHealth demonstrated improved data quality and timeliness in epidemic monitoring, prediction of population movement and infectious disease outbreaks, as well as effective monitoring of symptoms through patient care devices, remote vital sign monitors, and SMS messaging [60]. Besides, telehealth interventions in inaccessible areas significantly decreased the risk of death or loss to follow up. Additionally, eHealth interventions which often emerged as a response to changing security contexts in conflict-affected settings supported healthcare professionals' education, also addressed specific training challenges [56, 59, 60]. Bowsher et al. [59], however, highlighted disparities in the availability and development of eHealth tools, with eHealth initiatives predominantly originating from volunteer organisations on an ad hoc basis, lacking systematic protocols for design and service delivery.

In summary, FCAS suffer from a lack of health system tools, including physical resources, supervision, and data management. Supportive supervision, when undertaken appropriately,

is critical to deliver quality healthcare services. Interventions such as short-term training of managers and supervisors, practical tools, mentoring, and high-level support are helpful. Utilising eHealth interventions using standard protocols, systematic design, and delivery improve data management, quality, and timeliness. It also supports education and training in FCAS.

## II. Processes of care

Of the two components of this domain, 'competent care and systems', have more evidence, and aspects of 'positive user experience' were also included. Nine systematic reviews in various settings evidenced this domain.

In the review conducted by Jordan et al. [34], the major barriers of *'processes of care'* in conflict settings included poor quality healthcare characterised by incorrect diagnosis, inappropriate treatment of illnesses, inadequate patient referrals, lack of continuity of care, perception of judgmental or discriminatory behaviour from healthcare staff, language barriers, and insufficient communication. Regarding ***competent care and systems***, the quality-of-care dimensions were rarely addressed in the literature [52], and only Asgary et al. [37] discussed prevention strategies, specifically primary prevention interventions such as distribution of healthy food staples and lifestyle modifications, generally effective in contributing to reducing the incidence of NCDs. The review also identified common screening or prevention methods for NCDs including ultrasound screening and web-based technologies with varying degree of success in various humanitarian settings.

Asgary et al. [37] also proposed HSS initiatives that can collectively enhance the effectiveness of addressing NCDs within facility-based healthcare services. These initiatives, centred on capacity building, includes developing clinical guidelines, implementing health education, training staff, utilising pharmacy-level interventions, point-of-care laboratories, and eHealth tools. Similarly, Lassi et al. [36] emphasised the importance of improving resources for healthcare provider institutions and discussed strategies such as upgrading infrastructure and providing drugs and supplies within facility-based services in conflict-affected areas. The importance of capacity-building and preparing local health staff and patients to promote good clinical practice, monitor processes, and support patients' medication adherence and adaptability were also emphasised [36, 54]. Strengthening medication and supplies, particularly for managing NCDs in humanitarian settings was deemed critical [37]. Additionally, Ruby et al. [54] emphasised the inclusion of specific NCDs medications on essential medication lists to facilitate their accessibility and use during crisis situations. The authors also reported the success of algorithm-based interventions, as well as the benefits of monitoring individuals through electronic medical records, and systematically collecting baseline and routine NCDs data over time. Temporary settings, particularly camps with NGO-run primary healthcare services for displaced populations occasionally included basic referral structures to the public healthcare systems [52]. These services played critical roles in ensuring equitable access to health services as they were more accessible and affordable. However, challenges with coordination were observed due to the lack of understanding with the referral processes from primary healthcare to specialist services [52]. Healthcare workers, especially in conflict-affected areas, also demonstrated reluctance to refer patients with serious complications due to the dangerous roads to healthcare facilities [39]. Integration of healthcare services in certain fragile settings was non-standardised and often insufficient, failing to accommodate the frequent movement of patients across sectors and facilities [38, 52]. However, strategies such as combining childhood immunisations with other services including nutrition through a single access point, demonstrated improved uptake [44]. Yet, the success of integration has been showed to depend on various factors, such as the quality of the existing healthcare system and involvement of health and

non-health actors, including the local community and community leaders. Additionally, adapting national guidelines to the local context of healthcare management was suggested as crucial for achieving positive outcomes in the long run [38].

In terms of developing **positive user experience**, various mechanisms were discussed. These included improved physical accessibility, enhancing awareness of the need for specific healthcare services, building trust in service providers, and utilising multiple communication modes with target populations, such as through community leaders [44]. Using materials in local languages was helpful in making the public more receptive to health messages, thereby improving interactions with healthcare workers in various conflict-affected settings [42].

Overall, the evidence suggests addressing barriers to competent care and systems using strategies such as capacity building and enhancing resources within facility-based healthcare services, promoting integration of healthcare services and organised referral system, and prevention. Prioritising positive user experiences and adapting healthcare approaches to the local context are also very important. Implementing targeted interventions and involving key stakeholders, including local communities and leaders lead to better healthcare provision and patient experiences, ultimately strengthen the processes of healthcare delivery.

### III. Quality impacts

Of the three components of this domain, aspects of 'confidence in system' and 'economic benefit' were more evident, while the 'better health' component was the least reported. Eleven systematic reviews evidenced this domain.

In relation to **confidence in system**, trust emerged as key. Lack of confidence in the system, however, meant trust was often placed in local or traditional services particularly in fragile LMICs, even when urgent medical care was required. Homer et al. [39], for example, identified 'trust' both as crucial facilitator and obstacle to the delivery of midwifery care in humanitarian and fragile settings. Despite the availability of trained midwives, women opted for local untrained birth attendants due to the lack of communication, information sharing, or trust building with the community [39, 57]. Midwives also felt unsafe and unprotected through lack of trust, fearing that they might be held responsible for maternal deaths [39]. Schmid et al. [52] highlighted that, patients' trust in the health system influenced their health-seeking behaviour. Additionally, they found that, private sector facilities were often used as a primary access point for healthcare or alongside public healthcare services due to perceived poor quality of public healthcare system in a conflict-affected country, Iraq. In certain humanitarian settings, patients have expressed dissatisfaction with the low-quality healthcare they received, yet, in other instances their satisfaction remained even when the quality was poor [34]. Moreover, these patients sought healthcare services when the quality was reasonably good, even in challenging situations [41]. Besides, lack of available medicines also led to the use of private pharmacies during armed conflicts when public facilities faced shortages. Despite additional costs, however, patients continued to seek private-sector care, even when medicines availability improved, implying a lack of trust or confidence in the public healthcare system. As a result, building positive relationships, through partnership and respect, was suggested to improve communication links within health workers of all types and the community [57].

In terms of **economic benefit** of services, the reviews reported limited analysis of cost, due to the diversity of organisations providing healthcare, including charities and not-for-profit groups [37]. Political instability had a direct impact on the economic status of the public, influencing healthcare utilisation. Providing free or subsidised healthcare services and reducing out-of-pocket expenditure, including for refugee and internally displaced populations, successfully improved access, and healthcare utilisation by reducing financial burden and facilitating

financial risk protection [34, 41, 48]. Integrating services within existing systems was also suggested to help improving cost, scalability, or consistency in humanitarian settings [37].

According to Vivalya et al. [38], a higher risk of mental illnesses was reported in areas experiencing armed conflict and Ebola virus disease outbreaks. This was further complicated by the poor provision of healthcare, high rate of relapse, and a massive lack of needed services resulting in poor health outcomes [38]. Although not specifically designed to cover health expenses, cash transfers, whether conditional or unconditional, were reported as effective in promoting ***better health;*** they were used to improve nutrition, health, and education in displaced households. Additionally, cash transfers have demonstrated positive impacts on psychosocial and mental health and subjective well-being, often by alleviating the stress of financial burdens and promoting independence from aid [35, 51, 60]. van Daalen et al. [51] stated that, cash transfers were often preferred over other aid forms; and they have contributed to improving healthcare utilisation during the economic hardship caused by COVID-19, and increasing access to healthcare, and well-being of children aged under-five years. According to the review, conditional cash transfers were seen to be most effective when combined with health education intervention. Despite the positive impacts, however, social exclusion of recipients, verbal abuse from non-beneficiaries, and poor communication about logistics and timing of the transfers ending, were reported to cause stress and anxiety in participants [51].

Overall, the evidence suggests focusing on the quality of healthcare in disaster settings. Building trust in the health system is important to improve confidence and utilisation of formal healthcare services. Furthermore, integration of services, providing free or subsidised healthcare services and financial supports largely contribute to improve access to healthcare and health outcomes. Finally, Table 4 presents summaries of the identified challenges, interventions and outcomes of HSS within each topic.

## Discussion

This review presents the findings of systematic reviews on HSS across various FCAS contexts. The included studies employed diverse eligibility criteria, encompassing both generalised and specific aspects of health systems. Specific disaster situations, including conflict, disease outbreaks, and displaced populations, as well as broader contexts of FCAS or humanitarian settings, were examined in each paper. The findings of this review primarily concentrate on the 'foundations' of health systems, with limited data on 'process of care' and 'quality impacts,' highlighting that HSS initiatives and literature predominantly focus on inputs [30]. This review also shows that literature on HSS in FCAS predominantly focuses on challenges and interventions related to ongoing disaster situations and early recovery stages. Additionally, evidence suggests health interventions in FCAS are more short-term, primarily oriented towards providing humanitarian relief, lacking contributions to the broader development of health systems [61]. Witter et al. [62], however, suggested that literature on HSS, irrespective of fragility and conflict, is notably biased towards better-funded areas with increased external support and interest, possibly neglecting local-level innovations and smaller projects.

The importance of HSS is recognised globally for achieving universal health coverage and contributing to sustainable development goals [62, 63]. Despite being considered an abstract aim with challenging implementation, various approaches have been suggested to strengthen health systems in different settings [24, 30]. Notably, the WHO's policy directions on HSS in 2010 acknowledged a shift toward people-centred primary healthcare, placing people and communities at the centre [64]. This strategy is perceived to generate significant benefits, including improved access to healthcare, health outcomes, health literacy, job satisfaction among health workers, and service efficiency with reduced costs [65]. The present review

**Table 4. Summaries of challenges and interventions of HSS in FCAS.**

| Domains and components | Contexts | Challenges | Interventions/ Facilitators | Outcomes |
|---|---|---|---|---|
| **Population** | • Humanitarian crises in conflict, post-conflict or natural disasters settings in LMIC<br>• LMICs and wider humanitarian settings<br>• Settings with conflict-affected populations<br>• Countries listed as FCAS | Cultural and religious norms as barriers towards effective healthcare | • Empowering and engaging the community through community groups and leaders, refugees and other displaced populations<br>• Engaging and recruiting refugee healthcare workers in healthcare facilities<br>• Involving community-nominated volunteers in designing specific healthcare services and health promotion activities<br>• Community-driven multistakeholder programmes | • Local communities supported mitigating the impacts of crises<br>• Refugee healthcare workers supported the local workforce<br>• Local communities helped in financial contribution and resource mobilisation<br>• Community involvement in selecting individuals for community healthcare workers training increased their acceptance and facilitated their work<br>• Community involvement improved access, trust, and uptake of healthcare interventions<br>• Local communities played a role in mobilising political will for quality healthcare services |
| **Governance** | • Humanitarian, and mass displacement settings in LMICs<br>• Humanitarian crises during and after natural disasters and post-conflict<br>• Settings with conflict-affected populations<br>• Countries listed as FCAS | • Weak governance<br>• Unclear responsibilities, mistrust and tension<br>• Lack of and restrictive healthcare policies and implementation problems<br>• Lack of expertise and capacity<br>• Limited legal guidance<br>• Financial barriers: lack of inclusive finance systems, inefficient resource allocation, and a lack of clarity on financial needs<br>• Limited infrastructure<br>• Ineffective monitoring and evaluation<br>• Disorganised service delivery<br>• Fragmented/ uncoordinated health responses<br>• Irrelevant and proliferating new job descriptions without central coordination<br>• Donor's dominance of funding, decision-making and policymaking | • Political commitment<br>• Enhanced leadership<br>• Structural reforms<br>• People-centred governance<br>• Decentralisation<br>• Utilising various sources for information<br>• Health financing strategies such as community-based health insurance, performance-based financing, coordinated donor funding, and fees retention policy for local health facilities<br>• Reliable and cost-effective financial monitoring system with better coordination<br>• Management and directive coordination, such as the humanitarian cluster approach<br>• Using prior networks and infrastructure<br>• Cross-border governance and coordination initiatives to reduce the risk vaccine-preventable diseases transmissions<br>• Countries ownership of their health finance, in order to minimise the negative influences of external donors | • Governance improvements improved health system performance<br>• People-centred governance approach improved stakeholder engagement, accountability, setting a shared strategic direction, and stewarding resources<br>• Decentralisation supported health system reforms<br>• Health financing strategies facilitated health system reforms and services<br>• Cash transfers contributed to reducing economic hardships and poverty, increasing households' access to basic food, improving food security, or preventing acute malnutrition<br>• Management and directive coordination improved coordination amongst humanitarian actors<br>• Cluster system contributed to improving access to healthcare services and health outcomes<br>• Prior networks and infrastructure supported community health services, and emergency preparedness |

(*Continued*)

**Table 4.** (Continued)

| Domains and components | Contexts | Challenges | Interventions/ Facilitators | Outcomes |
|---|---|---|---|---|
| **Platforms** | • LMICs and wider humanitarian settings<br>• Conflict affected/ post-conflict settings<br>• Conflict, post-conflict or natural disaster settings in LMIC<br>• Armed conflict and Ebola Virus Disease outbreak<br>• Displaced populations in Iraq | • Contextual constraints and feasibility of HSS initiatives<br>• Barriers to implementation or success of new interventions<br>• Diminished and variable access and limited choices of healthcare platforms<br>• Health service interruptions<br>• Concerns with security and safety impacted health seeking behaviour | • Understanding local circumstances when designing healthcare models<br>• Strengthening public-private partnerships<br>• Using various healthcare models<br>• Establishing new clinics and supporting stationary clinics<br>• Central coordination and collaboration in community healthcare services<br>• Involving and building capacity of local community health workers<br>• Establishing a strong relationship between modern and traditional/ religious healers<br>• Coordinating with and recruiting traditional healthcare providers into the health system | • Mobile health services improved vaccination coverage<br>• Establishing new clinics, training healthcare workers, and supporting stationary clinics were effective in addressing NCDs<br>• Community health workers interventions improved access to healthcare<br>• Building the capacity of local community health workers and involving them in health promotion initiatives contributed to ensuring the continuity of essential healthcare services<br>• The use of traditional treatment in refugee settings improved health outcome |
| **Workforce** | • Humanitarian settings and emergencies<br>• Humanitarian crises in conflict, post-conflict or natural disaster settings in LMIC<br>• Settings with conflict-affected populations<br>• Conflict-affected/ post-conflict settings<br>• Active war<br>• Countries listed as FCAS | • Persistent workforce attrition/ flight and concentration in safer areas<br>• Inadequate salaries impacting quality of healthcare<br>• Oversaturated health labour market because of returnee healthcare workers<br>• NGOs and aid organisations contributing to labour market imbalances<br>• Lack of training and education opportunities<br>• Challenging provision and maintenance of healthcare professional education<br>• Low quality ad hoc or emergency on-the-job training<br>• Unregulated privatisation of training providers<br>• Inadequate educational quality assurance and standardisation<br>• Cultural norms, social situations and security concerns hindering recruitment and deployment/ transfer of the newly graduated healthcare workers, particularly female | • Programmes to reintegrate returned healthcare workers<br>• Expatriates recruitment<br>• Training healthcare workers in other countries to return to serve in the national public health sector<br>• Deployment of locally trained new graduates to rural and underserved areas as a post-graduation requirement<br>• Reduction of training duration<br>• Task shifting and training community health workers<br>• Adjusting education curricula to align with the context<br>• Enhancing national standards and organising medical bodies | • Recruitment of expatriate healthcare workers helped in providing short-term solutions<br>• Task shifting and training community health workers contributed to address shortages of trained and qualified healthcare workers |

(*Continued*)

**Table 4.** (Continued)

| Domains and components | Contexts | Challenges | Interventions/ Facilitators | Outcomes |
|---|---|---|---|---|
| **Tools** | • Populations affected by humanitarian crisis in LMICs<br>• Humanitarian settings and emergencies<br>• Humanitarian crises during and after natural disaster and post-conflict<br>• Settings with conflict-affected populations<br>• Conflict settings<br>• Active war<br>• Countries listed as FCAS | • Inadequate equipment, supplies, and infrastructure in clinical settings and training institutions<br>• Insufficient or lack of monitoring and supportive supervision<br>• Deficiency of capable managers and supervisors<br>• Poor-quality data, and inconsistent data collection | • Supportive supervision to identifying areas requiring development and capacity building<br>• Alternative means of supervision<br>• Short-term training, practical tools in a short course format, didactic training, on-site projects, mentoring,<br>• High-level support for managers and supervisors<br>• eHealth tools, such as mobile applications for information management | • Supportive supervision led to improved service delivery and performance of lay healthcare workers<br>• Remote supervision yielded similar results as in-person supervision<br>• Information and communication technologies supported health and medical care during disaster responses, and aided cash transfer programmes<br>• mHealth improved data quality and timeliness compared to paper-based methods<br>• Telehealth significantly decreased risk of death or loss to healthcare follow up in inaccessible areas<br>• eHealth supported healthcare professionals' education |
| **Processes of care** | • Populations affected by humanitarian crisis in LMICs<br>• Humanitarian and fragile settings<br>• Humanitarian setting in LMICs<br>• Displaced populations in Iraq<br>• Conflict-affected settings<br>• Armed conflict and Ebola Virus Disease outbreak | • Poor quality healthcare characterised by incorrect diagnosis, inappropriate treatment of illnesses, inadequate patient referrals, lack of continuity of care, perception of judgmental or discriminatory behaviour from healthcare staff, language barriers, and insufficient communication<br>• Challenges with coordination of healthcare services due to lack of understanding the referral processes<br>• Challenges to referral due to dangerous roads to healthcare facilities<br>• Non-standardised and often insufficient integration of healthcare services | • Screening and disease prevention strategies<br>• Clinical guidelines, health education, training staff, pharmacy-level interventions, point-of-care laboratories, upgrading infrastructure, drugs and supplies in facility-based services and eHealth tools<br>• Involving health and non-health actors, including the local community groups and leaders<br>• Adapting national guidelines to the local context<br>• Strategies such as improved physical accessibility, enhancing awareness of the need for specific healthcare services, building trust in service providers, and utilising multiple communication modes with populations | • Primary prevention interventions such as distribution of healthy food staples and lifestyle modifications contributed to reducing incidence of NCDs<br>• Screening or prevention methods for NCDs including ultrasound and web-based technologies was successful in varying degrees<br>• Strategies such as combining childhood immunisations with other services through a single access point improved uptake<br>• Community involvement improved integration of care<br>• Using materials in local languages was helpful in making the public more receptive to health messages |
| **Quality Impacts** | • Populations affected by humanitarian crisis and mass displacement in LMICs<br>• LMICs and wider humanitarian settings<br>• Humanitarian crises in conflict, post-conflict or natural disaster settings in LMIC<br>• Displaced populations in Iraq<br>• Armed conflict and Ebola Virus Disease outbreak<br>• Conflict/ war-affected countries | • Lack of communication, information sharing, or trust building with the community<br>• Perceived poor quality of public healthcare system, and variable quality of services<br>• Shortages of medicines in public facilities<br>• Political instability directly impact economic status of the public, influencing healthcare utilisation<br>• Lack and poor provision of healthcare resulting in poor health outcomes | • Building positive relationships, through partnership and respect<br>• Free or subsidised healthcare services and reduced out-of-pocket expenditure<br>• Integrating services within existing systems to help improving cost, scalability, or consistency<br>• Conditional and unconditional cash transfers | • Free or subsidised healthcare services and reduced out-of-pocket expenditure successfully improved access, and healthcare utilisation<br>• Cash transfers were effective in promoting better health, also improved nutrition, health, subjective well-being, and education<br>• Cash transfers contributed to improving healthcare utilisation and increasing access to healthcare |

indicated the crucial role of different community groups including refugee healthcare workers, traditional healthcare providers and other volunteers in strengthening the health system in FCAS beyond filling the workforce gaps. Local communities when placed at the centre can also support healthcare services provision and facilitate culturally appropriate interventions, leading to increased trust and healthcare utilisation. However, evidence on population needs and expectations is rare in the review.

According to the WHO, a well-functioning health system relies on trained and motivated health workers, well-maintained infrastructure, a reliable supply of medicines and technologies, adequate funding, robust health plans, and evidence-based policies [66]. However, many LMICs face deficiencies in basic resources such as finances, infrastructure, and workforce [24, 67]. The review indicates that fragility of states and conflict exacerbate these challenges, with unique issues such as targeted attacks on patients, healthcare workers, and infrastructure, unsafe roads, and a lack of healthcare workforce in insecure areas [16, 25, 61]. The included studies in this review also reported that, during armed conflicts or wars, health professional education and training suffer from a deficiency in expertise related to wartime topics; also due to reduced emphasis on civilian topics like primary and preventative care, coupled with a lack of standardisation and quality of the curriculum [56]. The interventions aimed at addressing these challenges, such as shortening training time and adapting the curriculum to the existing situation, while resolving immediate issues, may still have long-term repercussions. Equipment and infrastructure are also compromised due to targeted destruction and looting, and resources may divert to emergency responses, leaving regular healthcare services strained. As compared to stable states, strengthening the health system in FCAS has been more challenging, and the sustainability of implemented initiatives is questionable due to reasons such as the lack of a stable government and a higher possibility of relapse, particularly in contexts of armed conflicts [61]. These challenges may also relate to the different contexts and stages of FCAS. Witter et al. [62] identified a limited understanding of which HSS interventions and policies are most effective in specific contexts, particularly in conflict-affected countries and those undergoing transitions from aid or operating under distinct political arrangements. Likewise, evidence on the temporal and contextual differences, as well as long-term outcomes and impacts of individual interventions, is limited in our review which may also show that it takes long-time for the results to show. The review hence emphasises the importance of considering local circumstances when designing HSS interventions, given significant differences in stages and specific contexts even within the same situation. Understanding the impact of the crisis on health systems and considering the context and stages of fragility and conflict is also crucial for determining priorities in humanitarian responses [61, 68].

Disaster situations, including violent conflicts and disease outbreaks, pose significant threats to global health as their impacts often transcend national borders. Hence, effectively addressing the needs of patients across various settings necessitates cross-border coordination and diplomatic responses [69]. Moreover, during the initial stages of such disasters, national health systems may encounter challenges in coping independently. They may lack the capacity or willingness to coordinate responses efficiently and impartially [68]. Multiple humanitarian actors often get involved uncoordinated, leading to duplicated efforts with limited impact. The review identifies various coordination models globally, including the United Nations Humanitarian Cluster System, involving groups of humanitarian UN and non-UN organisations, which has proven to be effective in coordinating responses to large humanitarian emergencies [49, 70]. However, this system has been criticised for excluding local and national organisations from coordination activities and operating independently of the government in line with the humanitarian principle of 'independence' [71]. Besides, humanitarian responses such as the cluster system are meant to fill a temporary gap [70]. Karačić Zanetti et al. [69] also

highlighted criticism directed towards WHO and other international agencies for their delayed response in developing protocols and their lack of providing evidence-based and clear guidance to national health authorities. This deficiency was particularly noticeable during the early stages of the COVID-19 pandemic [69]. Nevertheless, our review underscores that HSS initiatives or programmes implemented independently of the national health system in FCAS may not be sustainable, unlike coordinated programmes within the national health system or integrated with existing infrastructure, which tend to be more effective and sustainable.

The limited evidence regarding the processes of care and quality impacts in this review may indicate the limitation in outcome measures and a diminished consideration of healthcare quality in FCAS. Quality is recognised as an intermediary objective in most national health policies, plans, and strategies, and strengthening the health system is considered a way to achieve quality healthcare services [63]. The quality of healthcare services is associated with public trust in the health system [25, 71, 72]. The review also recognises trust as an important factor in health-seeking behaviour and health outcomes [72]. Kruk et al. [24] in their systematic analysis of amenable deaths, reported the detrimental effect of low-quality health systems on preventable mortality in LMICs. Nevertheless, a limited number of studies have established a connection between HSS interventions and health outcomes. Furthermore, these studies commonly do not employ health outcomes as measures for evaluating HSS interventions, resulting in heterogeneity when considered [62].

As the present review also indicates, evaluation criteria in health systems typically concentrate on inputs such as tools and workforce, although drawing attention towards the manifestations of quality, such as acceptability, cultural appropriateness, and responsiveness has been suggested beyond technical measures [30, 73]. High-quality health information is required for performance indicators and assessing healthcare quality, and health information systems in countries across all income levels fall short of their potential to contribute to improving health system performance. Data on healthcare quality in health information systems are often limited and inconsistently documented in paper-based records in many low-resource settings [74]. The review also finds FCAS being characterised by poor-quality data, and difficulty of data collection due to safety concerns, lack of resources, and contextual problems despite promising evidence on digital technology [75]. Besides, the volatility of disaster situations can hinder monitoring and evaluation [66, 75]. The dynamic nature of crises and recovery across different stages may further complicate tracking progress and evaluation. This necessitates considering alternative means of data collection or monitoring and evaluation approaches for health systems in FCAS. The challenges associated with the volatility and unpredictability of situations, as well as learning from and adapting to the changing context, also need to be considered [76, 77].

This study, despite its rich variety of data, has limitations that need to be acknowledged. *Firstly*, the included literature in this review is heterogeneous and cover various areas of health system and contexts making synthesis challenging. However, the influence of shocks and coping strategies is similar across different crisis situations, including conflict, epidemics, and other natural disasters that may all result in humanitarian crises [12, 55] rationalising the synthesis, and ultimately the review. Nonetheless, the stages and specific contexts even within the same situation can significantly differ. *Secondly*, the systematic reviews also included literature from relatively stable countries mainly due to fragile contexts such as refugees and displaced populations. Consequently, the initiatives and strategies discussed in the reviews also include those in fragile contexts within otherwise stable states. *Thirdly*, this review is limited in providing complete depiction of HSS, as the evidence predominantly focuses on initiatives within or shortly after disaster situations and the initial stages of recovery. As nations progress in certain areas such as peace restoration and economic development beyond the aftermath of disasters

or crises, they may cease to be classified as FCAS. Despite this progress, health systems may continue to struggle with lingering effects, necessitating sustained, long-term efforts for strengthening. Unfortunately, existing studies may not explicitly address this post-crisis context as FCAS. *Fourthly*, given that the concept of 'health systems' remains somewhat indistinct, it has presented challenges in delineating clear boundaries to establish eligibility criteria for our review. As a result, some potentially relevant studies may have been inadvertently excluded from consideration. *Lastly*, only published, and English language literature have been included. Thus, relevant data that could enhance the study may have been missed. However, this review used extensive search terms and databases as to make the search as robust as possible. The findings have also been tailored to the utilised conceptual framework exhaustively.

## Conclusions

This comprehensive review of systematic reviews demonstrates the complex and multifactorial challenges faced by health systems in FCAS, notably, poor access and compromised quality of healthcare due to the lack of essential inputs such as workforce, tools, and effective governance systems. The concentration of evidence on the 'foundation' aspects of health system indicates a predominant focus on input centric HSS initiatives. While health systems encounter similar challenges across various settings and contexts, FCAS presents distinct obstacles contributing unique dimensions to the problem, further intensifying the challenges.

Given the multifaceted nature of HSS, a holistic approach with adequate resources and stakeholder's coordinated effort is essential. It is also important for global health actors and international development/aid organisations to shift their focus towards strengthening health systems in resource-limited settings and FCAS, considering the actual needs aligned with the countries priorities. Beyond technical and material support, stakeholders need to prioritise the processes and outcomes of health system functions, including the capacity and quality of healthcare services. Furthermore, the present review suggests the importance of a nuanced understanding of the diverse challenges across FCAS, advocating for flexible and adaptable HSS interventions tailored to specific contexts, needs, and stages of crises.

## Supporting information

**S1 Annex. Search strategy utilised for database searching.**
(DOCX)

**S2 Annex. Summary of the main findings for the reviewed papers.**
(DOCX)

**S3 Annex. Completed PRISMA checklist.**
(DOCX)

**S4 Annex. Methodological quality appraisal using the JBI tool.**
(DOCX)

## Acknowledgments

We would like to acknowledge King's College London, London, UK and St Paul's Hospital Millennium Medical College, Addis Ababa, Ethiopia.

We would also like to acknowledge Professor Funmi Olonisakin, for her support in the PhD programme; and Sara Montalti, for her guidance in developing an effective search strategy, and online database searching.

## Author Contributions

**Conceptualization:** Birke Bogale, Sasha Scambler, Jennifer E. Gallagher.

**Data curation:** Birke Bogale, Sasha Scambler, Aina Najwa Mohd Khairuddin, Jennifer E. Gallagher.

**Formal analysis:** Birke Bogale.

**Investigation:** Birke Bogale, Sasha Scambler, Aina Najwa Mohd Khairuddin, Jennifer E. Gallagher.

**Methodology:** Birke Bogale, Sasha Scambler, Aina Najwa Mohd Khairuddin, Jennifer E. Gallagher.

**Project administration:** Birke Bogale.

**Resources:** Birke Bogale, Jennifer E. Gallagher.

**Software:** Birke Bogale, Aina Najwa Mohd Khairuddin.

**Supervision:** Sasha Scambler, Jennifer E. Gallagher.

**Validation:** Sasha Scambler, Aina Najwa Mohd Khairuddin, Jennifer E. Gallagher.

**Visualization:** Birke Bogale, Sasha Scambler, Aina Najwa Mohd Khairuddin, Jennifer E. Gallagher.

**Writing – original draft:** Birke Bogale.

**Writing – review & editing:** Birke Bogale, Sasha Scambler, Aina Najwa Mohd Khairuddin, Jennifer E. Gallagher.

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
