## [Decision Letter · Decision Letter 0]

2 Jan 2024

PONE-D-23-25820Health system strengthening in fragile and conflict-affected states: a review of systematic reviewsPLOS ONE

Dear Dr. Birke Bogale,

Thank you for submitting your manuscript to PLOS ONE. After careful consideration, we feel that it has merit but does not fully meet PLOS ONE’s publication criteria as it currently stands. Therefore, we invite you to submit a revised version of the manuscript that addresses the points raised during the review process.

**ACADEMIC EDITOR: **Make the required changes suggested in your Results section. Clarify the FCAS definition, elaborate on the bibliometrics section, quality appraisal in the main text of the manuscript. Also, add a Table in an online annex detailing the quality appraisal as suggested.Clarify in detail what criteria were utilized to decide which studies were eligible to be included in the review. This can be done in the Methods section. Also, justify why Jaung et al ( see doi: 10.1093/heapol/czab007 ) not eligible when it seems much more related to health systems than Ruby et al. as identified by the reviewer. Please address the comments and suggestions on your Discussion and Limitations section. Rewriting the discussion section is recommended given the reviewers feedback. A thorough proofreading for minor errors and grammatical mistakes are recommended. ==============================

We look forward to receiving your revised manuscript.

Kind regards,

Rushdia Ahmed, MPH, MA

Academic Editor

PLOS ONE

Journal Requirements:

"We would like to acknowledge King’s College London, London, UK for fully funding the lead author’s (BB) study for Dental and Health Sciences Research MPhil/PhD; also, St Paul’s Hospital Millennium Medical College, Addis Ababa, Ethiopia.

We would also like to acknowledge Professor Funmi Olonisakin, for her support in the PhD programme; and Sara Montalti, for her guidance in developing an effective search strategy, and online database searching in this study."

"BB is funded by the King’s College London ‘Africa International PGR Scholarships 2021-22’ (URL: https://www.kcl.ac.uk/study-legacy/funding/africa-international-pgr-scholarships) to support her study ‘Dental and Health Sciences Research MPhil/PhD’.

JEG and SS are salaried by King’s College London.

The funder had no role in study design, data collection and analysis, decision to publish, or preparation of the manuscript."

Reviewers' comments:

Reviewer's Responses to Questions

**Comments to the Author**

1. Is the manuscript technically sound, and do the data support the conclusions?

Reviewer #1: Yes

Reviewer #2: Yes

2. Has the statistical analysis been performed appropriately and rigorously? 

Reviewer #1: N/A

Reviewer #2: N/A

3. Have the authors made all data underlying the findings in their manuscript fully available?

Reviewer #1: Yes

Reviewer #2: Yes

4. Is the manuscript presented in an intelligible fashion and written in standard English?

Reviewer #1: Yes

Reviewer #2: Yes

5. Review Comments to the Author

Reviewer #1: This review of systematic reviews sought to o explore health system strengthening in fragile and conflict-affected states and synthesize the evidence from published literature. It addresses an important topic. The design is appropriate, and it is well conducted and nicely written. The conceptual framework is appropriate and helpful.

Results:

I find some of the results a bit odd. For example, the Ruby review being included when none of the papers in Ruby’s review seemed specifically about health systems? Related to this, why was a review on models of NCD care in humanitarian crises by Jaung et al ( see doi: 10.1093/heapol/czab007 ) not eligible when it seems much more related to health systems than Ruby et al.? I can also think of many systematic reviews on other health outcomes (e.g. mental health interventions among conflict-affected populations) that would appear as relevant as Ruby to health systems but were not considered eligible for this review. For example, https://gh.bmj.com/content/4/5/e001484.info and https://doi.org/10.1186/s13033-020-00431-1. I appreciate there’s a conceptual fuzziness with health systems that can make it difficult to determine inclusion/exclusion criteria of individual papers but I think this needs better explanation. It may also be worth noting this fuzziness as a potential limitation?

There was little information on how the findings from the eligible reviews relate to different phases of crisis (i.e. acute, chronic, prolonged, post-conflict/crisis rebuilding). These temporal aspects related to health systems are really important and more investigation would have been helpful. Could the findings in relation to different phases be brought out more?

No results on quality appraisal are given apart from in Table 3. They should be given as summarized form in the main text of the manuscript and then in more detail using a Table in an online annex.

Discussion:

The Discussion section is generally quite nicely written. However, it is quite repetitive of the Results section (including re-using a lot of the citations from the Results section). It would be more helpful if the Discussion section could put the findings in a broader context – for example, how it relates humanitarian guidelines and policies, findings of reviews of health systems in stable LMIC settings.

Limitations section:

I found most of the Limitations section quite impenetrable.

For example, in the second limitation (which is a valid and important one), the first sentence doesn’t seem to make any sense (“the systematic reviews included literature from countries with a history of being classified as FCAS or not, confirming the possibility of fragility manifesting within any setting or system”). What does this actually mean? Perhaps it could be more clearly written.

For example, I didn’t understand the third limitation (“this review may have only included HSS initiatives focused on the earlier stages of crises and may lack coverage of all aspects or the later stages of HSS of FCAS”). Shouldn’t the authors be able to determine this by looking at the papers, rather than just speculate on it?

For example, I also didn’t understand this. “Fragile states may also mean LMICs because of the complexity of the fragile situations which slows down their progress”. How is this a limitation? All fragile states are LMICs aren’t they? It’s also poorly written.

I also didn’t get why “the Lancet’s high-quality health system framework may have

indirectly targeted FCAS”. Why would it have done this and why is this a limitation?

Reviewer #2: This is a very interesting and comprehensive review of reviews, however, the manuscript could be improved substantially.

Minor comments

Abstract: you note the importance of HSS and quality- but do not define quality of what - though I can see that later you reference the high-quality HS framework.

Line 62 introduction: humanitarian crises are not just disasters - remove the bracket

Line 96 - Assume you mean Figure 1

Line 109 - PRISMA is a reporting guideline, not a ‘how to do a systematic review’ guideline, please correct

Line 233 - do you mean lack of accountability?

Line 250 - do you mean decentralization supported health system reform? This needs to be unpacked a bit, not clear

Lines 618-629 - If I understand correctly, you wish to say: findings of included reviews predominantly focus on inputs but neglect to focus on processes/outcomes etc. The first 2 lines are confusing and can be cut in the paragraph, and the rest really needs to be reworked - and ideally contextualized: do the included systematic reviews present their data according to WHO building blocks? What framings do they use?

Lines 635 - what framework?

Lines 638-640 seem like a big over generalization. My review stops here.

Major comments

A more thorough bibliometrics section should be included at front - there is too little here - how many of the countries included fall into what type of FCAS definition? What are the broader problems that these studies consider? What is the nature of the fragility discussed etc.

For each of your main sections please clearly list how many reviews considered that topic

Governance section: more broadly, this section and others would benefit from a figure or diagram where you clearly identify main challenges and HSS approaches that have worked. The text here is quite descriptive and dry, which is understandable, but at the minute it is very difficult to follow

My review stopped mid-way through discussion as there are quite sweeping statements made here that do not always link back clearly to your findings. Across your presentation of findings there is a tendency to treat all FCAS as a uniform group - this is actually quite damaging. I.e. active conflict is very different to past conflict, which in turn is very different to a localized natural disaster or to persistent climatic change (e.g. repeated drought). Was there any analysis done to unpack any potential differences in findings according to the diverse nature of the FCAS in question? The health workforce section is the only one that teases out these differences, so maybe try that for the other sections too. How do the reviews you included deal with this heterogeneity with the FCAS? What definitions did they employ for inclusion/exclusion?

6. PLOS authors have the option to publish the peer review history of their article (what does this mean?). If published, this will include your full peer review and any attached files.

Reviewer #1: No

Reviewer #2: **Yes: **Karin Diaconu

---

## [Author Response · Author response to Decision Letter 0]

17 Feb 2024

Reviewer #1: This review of systematic reviews sought to explore health system strengthening in fragile and conflict-affected states and synthesize the evidence from published literature. It addresses an important topic. The design is appropriate, and it is well conducted and nicely written. The conceptual framework is appropriate and helpful.

Author’s Response (AR): Thank you.

Results:

I find some of the results a bit odd. For example, the Ruby review being included when none of the papers in Ruby’s review seemed specifically about health systems? Related to this, why was a review on models of NCD care in humanitarian crises by Jaung et al ( see doi: 10.1093/heapol/czab007 ) not eligible when it seems much more related to health systems than Ruby et al.? I can also think of many systematic reviews on other health outcomes (e.g. mental health interventions among conflict-affected populations) that would appear as relevant as Ruby to health systems but were not considered eligible for this review. For example, https://gh.bmj.com/content/4/5/e001484.info and https://doi.org/10.1186/s13033-020-00431-1. I appreciate there’s a conceptual fuzziness with health systems that can make it difficult to determine inclusion/exclusion criteria of individual papers but I think this needs better explanation. It may also be worth noting this fuzziness as a potential limitation?

AR: We aimed to look at health system strengthening in its broad form, and the utilised framework contributed to our eligibility criteria. We excluded papers on specific health interventions or those solely targeting provisions of care for individual health conditions rather than more generalised/wider approaches. We also excluded those targeting specific population groups, such as age groups, and professions.

We agree that the studies included in Jaung et al., (2021) are related to health system and its inclusion may have added value to the findings of our review. However, the subject of this paper, ‘models of care specific to patients with diabetes and hypertension’ was regarded as too condition specific. Our inclusion/exclusion criteria excluded these condition specific papers to enable us to look at papers that looked at health system strengthening more broadly. 

There was little information on how the findings from the eligible reviews relate to different phases of crisis (i.e. acute, chronic, prolonged, post-conflict/crisis rebuilding). These temporal aspects related to health systems are really important and more investigation would have been helpful. Could the findings in relation to different phases be brought out more?

AR: The evidence on the temporal differences of the findings in the included systematic reviews is very limited. The systematic reviews mostly did not consider the temporality, and some papers acknowledged this as a limitation. However, we tried to integrate this aspect in the review where the information was available. We have also provided more explanations in the response to reviewer 2.

No results on quality appraisal are given apart from in Table 3. They should be given as summarized form in the main text of the manuscript and then in more detail using a Table in an online annex.

AR: We have included a summary in the results section, also added more detail. An online annex (S4 Annex) of the detail has also been provided.

Discussion:

The Discussion section is generally quite nicely written. However, it is quite repetitive of the Results section (including re-using a lot of the citations from the Results section). It would be more helpful if the Discussion section could put the findings in a broader context – for example, how it relates humanitarian guidelines and policies, findings of reviews of health systems in stable LMIC settings.

AR: Thank you, this is really helpful, we have rewritten the discussion section.

Limitations section:

I found most of the Limitations section quite impenetrable.

For example, in the second limitation (which is a valid and important one), the first sentence doesn’t seem to make any sense (“the systematic reviews included literature from countries with a history of being classified as FCAS or not, confirming the possibility of fragility manifesting within any setting or system”). What does this actually mean? Perhaps it could be more clearly written.

For example, I didn’t understand the third limitation (“this review may have only included HSS initiatives focused on the earlier stages of crises and may lack coverage of all aspects or the later stages of HSS of FCAS”). Shouldn’t the authors be able to determine this by looking at the papers, rather than just speculate on it?

For example, I also didn’t understand this. “Fragile states may also mean LMICs because of the complexity of the fragile situations which slows down their progress”. How is this a limitation? All fragile states are LMICs aren’t they? It’s also poorly written.

I also didn’t get why “the Lancet’s high-quality health system framework may have indirectly targeted FCAS”. Why would it have done this and why is this a limitation?

AR: Thank you. This is very helpful; the section has now been rewritten to make it clearer.

 

Reviewer #2: This is a very interesting and comprehensive review of reviews, however, the manuscript could be improved substantially.

Minor comments

Abstract: you note the importance of HSS and quality- but do not define quality of what - though I can see that later you reference the high-quality HS framework.

AR: Thank you, we have specified it as ‘quality in health systems’ for more clarity.

Line 62 introduction: humanitarian crises are not just disasters - remove the bracket

AR: The term ‘disaster’ in the specified section intended to show that the terms humanitarian crisis and humanitarian disasters are used in the literature interchangeably. However, we have removed the bracket as per your suggestion. 

Line 96 - Assume you mean Figure 1

AR: Thank you for pointing this out, it is now corrected as per the order of appearance. We have also resubmitted the diagram that appears in the manuscript first renamed as ‘Fig 1’.

Line 109 - PRISMA is a reporting guideline, not a ‘how to do a systematic review’ guideline, please correct

 AR: Thank you. It is corrected.

Line 233 - do you mean lack of accountability?

 AR: ‘Perceived lack of accountability’, explains it better. It is changed, thank you.

Line 250 - do you mean decentralization supported health system reform? This needs to be unpacked a bit, not clear

 AR: Yes, it is changed to ‘decentralisation supported health system reform’. 

Lines 618-629 - If I understand correctly, you wish to say: findings of included reviews predominantly focus on inputs but neglect to focus on processes/outcomes etc. The first 2 lines are confusing and can be cut in the paragraph, and the rest really needs to be reworked - and ideally contextualized: do the included systematic reviews present their data according to WHO building blocks? What framings do they use?

AR: Thank you. The discussion section is rewritten. We did not include the framings the systematic reviews have used since they studied different areas of health system and follow different approaches; also, not all considered using available health system frameworks.

Lines 635 - what framework?

 AR: It is specified as ‘the high-quality health system framework’.

Lines 638-640 seem like a big over generalization. My review stops here.

AR: It was based on existing literature, which may not be explicitly written. We have removed it in the current draft.

Major comments

A more thorough bibliometrics section should be included at front - there is too little here - how many of the countries included fall into what type of FCAS definition? What are the broader problems that these studies consider? What is the nature of the fragility discussed etc.

AR: We have added more details in the bibliometric section, including the details of the countries in terms of the world bank’s list of fragile and conflict-affected situations. However, we only manually extracted the countries mentioned in the systematic reviews. Our list of countries may not represent all the studied countries, as the systematic reviews also used collective names/terms and different ways of reporting.

For each of your main sections please clearly list how many reviews considered that topic

 AR: We have included the number of reviews in each section.

Governance section: more broadly, this section and others would benefit from a figure or diagram where you clearly identify main challenges and HSS approaches that have worked. The text here is quite descriptive and dry, which is understandable, but at the minute it is very difficult to follow.

AR: Thank you for the comment, we believe the key findings/ HSS initiatives column in ‘Table 3’ and ‘S2 Annex’ containing summaries of the main findings helps in following the results.

We have updated the results section and the summary of the ‘governance’ section to make it easier to follow. We have also included an additional table (Table 4) with the specified challenges, and interventions that were reported highlighted.

My review stopped mid-way through discussion as there are quite sweeping statements made here that do not always link back clearly to your findings. Across your presentation of findings there is a tendency to treat all FCAS as a uniform group - this is actually quite damaging. I.e. active conflict is very different to past conflict, which in turn is very different to a localized natural disaster or to persistent climatic change (e.g. repeated drought). Was there any analysis done to unpack any potential differences in findings according to the diverse nature of the FCAS in question? The health workforce section is the only one that teases out these differences, so maybe try that for the other sections too. How do the reviews you included deal with this heterogeneity with the FCAS? What definitions did they employ for inclusion/exclusion?

AR: In this paper we attempted to map out the findings in terms of the stages and contexts of the situation. The systematic reviews included in this paper vary in the types of fragility and the elements of health system strengthening addressed, as well as the diverse criteria used in the included systematic reviews. For example, some papers utilised ‘humanitarian setting/crisis’ to describe collective contexts or settings such as conflict/post-conflict, natural disasters, and displaced population/refugees, while others described them in terms of FCAS and synthesised their data accordingly. In addition, while some papers presented the findings in terms of specific contexts such as conflict-affected, post-conflict or displaced populations, others were described collectively as humanitarian or fragile contexts. The temporality of the interventions is also not addressed in most reviews. However, we have included all reviews studying contexts related to FCAS that are specified in the paper regardless of their eligibility criteria, and the definitions or metrics used. Narrowing down the inclusion criteria may have led to losing relevant information/evidence.

Most of the evidence related to the workforce came from ‘conflict-affected’ contexts (during or post-conflict) and involves relatively homogenous or interconnected subject areas unlike all the other topics. Overall, our presentation of the findings has been shaped by how the included systematic reviews treated and categorised the countries/contexts included in their papers and how they presented their findings.

---

## [Decision Letter · Decision Letter 1]

6 May 2024

PONE-D-23-25820R1Health system strengthening in fragile and conflict-affected states: a review of systematic reviewsPLOS ONE

 Subject: Revision and Further Suggestions for Manuscript Improvement

Dear Dr. Bogale,

Thank you for submitting the revised version of your manuscript. We appreciate your efforts in addressing the reviewers' comments and suggestions. Your dedication to enhancing the quality of the manuscript is commendable.

We have carefully reviewed the revised manuscript and are pleased to see the improvements made based on the reviewers' feedback. However, there is one comment from Reviewer 1 that we believe could further strengthen the manuscript.

Moreover, we would like to draw your attention to the potential significance of diplomatic responses in strengthening health systems, as mentioned in the article "Diplomatic response to global health challenges in recognizing patient needs: A qualitative interview study" (https://www.frontiersin.org/journals/public-health/articles/10.3389/fpubh.2023.1164940/full). Diplomatic responses play a crucial role in addressing global health challenges and ensuring that patient needs are recognized and met effectively. We encourage you to consider integrating insights from this study into your manuscript to enrich the discussion on strategies for enhancing health system resilience during crises.

We kindly request that you incorporate the suggested revisions and send the revised manuscript back to us at your Jun 20 2024 11:59PM. We are confident that these additions will further enhance the value and impact of your work.

Thank you once again for your commitment to advancing knowledge in this important field. We look forward to receiving the revised manuscript.

Reviewers' comments:

Reviewer's Responses to Questions

**Comments to the Author**

1. If the authors have adequately addressed your comments raised in a previous round of review and you feel that this manuscript is now acceptable for publication, you may indicate that here to bypass the “Comments to the Author” section, enter your conflict of interest statement in the “Confidential to Editor” section, and submit your "Accept" recommendation.

Reviewer #1: (No Response)

Reviewer #2: All comments have been addressed

2. Is the manuscript technically sound, and do the data support the conclusions?

Reviewer #1: Yes

Reviewer #2: Yes

3. Has the statistical analysis been performed appropriately and rigorously? 

Reviewer #1: N/A

Reviewer #2: N/A

4. Have the authors made all data underlying the findings in their manuscript fully available?

Reviewer #1: Yes

Reviewer #2: Yes

5. Is the manuscript presented in an intelligible fashion and written in standard English?

Reviewer #1: Yes

Reviewer #2: Yes

6. Review Comments to the Author

Reviewer #1: Thank you for the changes. I'm happy with most of them. However, I still don't understand how a review on interventions for NCDs (Ruby et al) is eligible, but a review of models of care on NCDs is not eligible. Both reviews were of studies examining individual types of NCDs (e.g. diabetes, hypertension etc). Indeed, the review of models of care seems much more relevant to health systems than Ruby's et al of specific interventions addressing NCDs (none of which seems to be about health systems). Your response does not seem very plausible. At the very least you could acknowledge in the Limitations section the challenges of defining boundaries for your review of reviews and that certain potentially eligible reviews may have been inadvertently excluded.

Reviewer #2: (No Response)

7. PLOS authors have the option to publish the peer review history of their article (what does this mean?). If published, this will include your full peer review and any attached files.

Reviewer #1: No

Reviewer #2: **Yes: **Karin-Daniela Diaconu

---

## [Author Response · Author response to Decision Letter 1]

10 May 2024

We would like to thank both the reviewers and the editor for reviewing our manuscript and providing very constructive comments and suggestions.

Reviewer 1: Many thanks for the comment, we agree that it needs to be put as a limitation and we have included it the revised manuscript.

Editor: Many thanks for suggesting the article on 'diplomatic responses'. It is very useful, and we have integrated insights from the paper into the discussion section of our manuscript.

---

## [Decision Letter · Decision Letter 2]

28 May 2024

Health system strengthening in fragile and conflict-affected states: a review of systematic reviews

PONE-D-23-25820R2

Dear authors,

I am pleased to inform you that, after thorough review and consideration, your manuscript titled "[Health system strengthening in fragile and conflict-affected states: a review of systematic reviews

]" has been accepted for publication .

Your dedication and effort in addressing the reviewers' comments and revising the manuscript have significantly enhanced the quality and impact of your work.

We look forward to the successful publication of your work and your continued contributions to our journal.

Reviewers' comments:

Reviewer's Responses to Questions

**Comments to the Author**

1. If the authors have adequately addressed your comments raised in a previous round of review and you feel that this manuscript is now acceptable for publication, you may indicate that here to bypass the “Comments to the Author” section, enter your conflict of interest statement in the “Confidential to Editor” section, and submit your "Accept" recommendation.

Reviewer #3: (No Response)

Reviewer #4: (No Response)

2. Is the manuscript technically sound, and do the data support the conclusions?

Reviewer #3: Yes

Reviewer #4: Yes

3. Has the statistical analysis been performed appropriately and rigorously? 

Reviewer #3: N/A

Reviewer #4: N/A

4. Have the authors made all data underlying the findings in their manuscript fully available?

Reviewer #3: Yes

Reviewer #4: Yes

5. Is the manuscript presented in an intelligible fashion and written in standard English?

Reviewer #3: Yes

Reviewer #4: Yes

6. Review Comments to the Author

Reviewer #3: This is a very important topic and the authors have done a good job in synthesizing the wide array of evidence while adhering to a sound methodology and ensuring rigour.

Reviewer #4: The paper is a peace of art , deserve definitely to published, congratulation a=in advance for all of the great team

It will make a huge contribution to the HSS literature

I enjoyed each and every paragraph read it once and twice, It seems that it has been extremely reviewed before and benefited a lot from other comments

7. PLOS authors have the option to publish the peer review history of their article (what does this mean?). If published, this will include your full peer review and any attached files.

Reviewer #3: No

Reviewer #4: **Yes: **Maher ALAREF

---

## [Editor Report · Acceptance letter]

5 Jun 2024

PONE-D-23-25820R2 

PLOS ONE

Dear Dr. Bogale, 

I'm pleased to inform you that your manuscript has been deemed suitable for publication in PLOS ONE. Congratulations! Your manuscript is now being handed over to our production team.

Kind regards, 

on behalf of

Dr. Jasna Karacic Zanetti 

Academic Editor

PLOS ONE